



# Reduced transpiration without changes in root water uptake patterns in degraded trees in semi-arid afforestation ecosystems

Junjie Dai[1,2,3], Ying Zhao[1,2,3], Katsutoshi Seki[4], Li Wang[1,2,5]

[1]State Key Laboratory of Soil Erosion and Dryland Farming on the Loess Plateau, the Research Center of Soil and Water

Conservation and Ecological Environment, Chinese Academy of Sciences and Ministry of Education, Yangling, Shaanxi 712100, China

[2]Institute of Soil and Water Conservation, Chinese Academy of Sciences and Ministry of Water Resources, Yangling, Shaanxi 712100, China

[3]University of Chinese Academy of Sciences, Beijing 100049, China

[4]Natural Science Laboratory, Toyo University, Bunkyo-ku, Tokyo 112-8606, Japan

[5]Institute of Soil and Water Conservation, Northwest A&F University, Yangling 712100, China

*Correspondence to*: Li Wang (wangli5208@nwsuaf.edu.cn)

**Abstract.** While reforestation in water-limited areas has increased vegetation coverage, an increasing number of studies have reported that large-scale plantations are suffering from tree degradation, which is characterized by short trees, small size, and

dieback. Moreover, the changes in hydrological processes of degraded trees within the soil-plant system remain poorly understood. Here, the stable isotope method ($^2$H, $^{18}$O, and $^{13}$C) and thermal dissipation technique were used to compare the soil water dynamics, tree transpiration, root water uptake patterns, and intrinsic water-use efficiency (WUE$_i$) of *Populus simonii* under different degradation degrees (no degradation, ND; light degradation, LD; or severe degradation, SD) in the 2021 growing season. As tree degradation intensified, the root weight density decreased significantly ($P<0.05$) and the root

proportion of the shallow layer (0–40 cm) increased. Influenced by precipitation recharge, the soil volumetric water content (SWC) of the shallow layer did not show significant differences ($P>0.05$) among the different degradation degrees. The SWC of the middle (40–80 cm) and deep (80–200 cm) layers were significantly ($P<0.05$) lower in the ND plot than the LD and SD plots. Despite the differences in SWC, the isotopic characteristics of the soil water and xylem water were similar among the ND, LD, and SD plots. Although tree transpiration in the LD and SD plots was significantly reduced ($P<0.05$)

compared to that in the ND plot, the main depths and proportions of water uptake by the root system did not change. *P. simonii* trees in the ND, LD, and SD plots were able to shift the water source from shallow to deep layers in the process of soil wetting to drying. Moreover, compared to healthy trees, WUE$_i$ of degraded trees was more sensitive to SWC. Our study shows that although degraded trees alleviate the exhaustion of deep soil water reservoirs to some extent, the sustainable development of afforestation ecosystems requires appropriate stand management measures to maintain balanced forest-water

relationships.



# 1 Introduction

Forests account for approximately 45% of global terrestrial carbon storage, play an essential role in the water cycle of terrestrial ecosystems, and provide various critical ecosystem services for maintaining biodiversity (Choat et al., 2018; Liu et al., 2021a). However, global warming has caused more frequent regional drought events and increased the duration and intensity of droughts (Allen et al., 2010), especially in arid and semi-arid areas (Iqbal et al., 2021). These changes led to lower water availability and soil degradation in forests and reduced biodiversity in the affected areas, thereby reducing the forest area. To increase the carbon sink, enhance biodiversity, and prevent wind and soil erosion in water-limited areas, a series of afforestation ecological engineering projects have been implemented worldwide in recent decades (Chen et al., 2019; Yang et al., 2022), especially in China (Guo, 2021; Su and Shangguan, 2019). Although China only accounts for 6.6% of the vegetation area worldwide, it contributed 25% of the global net increase in leaf area from 2000 to 2017, thus contributing to global greening efforts (Chen et al., 2019).

Generally, the species used in afforestation in water-limited areas have strong drought adaptability and can regulate water-use strategies under changing water conditions. For instance, drought-tolerant species can shift the water source from the shallow soil layer to deep soil layer through dimorphic roots (Wang et al., 2017), increase the intrinsic water-use efficiency ($WUE_i$) (Wang et al., 2021), or display more isohydric behavior to reduce water loss (Ding et al., 2021) under water stress conditions. However, an increasing number of studies of afforestation projects have reported the emergence of numerous degraded trees, which are characterized by short height, small size, and dieback (Chen et al., 2015; Liu et al., 2020; McVicar et al., 2010; Sun et al., 2018; Zhang et al., 2020). Compared to the functions of normal-growing trees, those of degraded trees declined significantly. Degraded trees might present a lower transpiration rate due to the small tree size and narrower canopy width, and they might also lose the ability to utilize deep soil water from the lack of deep root systems (Liu et al., 2021a). Because the availability of deep-water sources often enables trees to adopt isohydric behavior to mitigate drought stress (Brinkmann et al., 2016; Ding et al., 2021), degraded trees with low deep-root activity may be vulnerable to drought (Liu et al., 2021a). Currently, most studies suggest that tree degradation or mortality is associated with water stress (Liu et al., 2021a; Sun et al., 2018; Zhang et al., 2020). Under drought conditions, trees often suffer from reduced photosynthetic carbon sequestration and non-structural carbohydrate depletion (McDowell et al., 2008), which might stunt tree growth. Therefore, the changes in hydrological processes within the soil-plant system between degraded trees and healthy trees must be clarified. As techniques for tracing stable isotopes ($^2$H, $^{18}$O, and $^{13}$C) have matured, the methods have been extensively used in critical eco-hydrology topics, such as determining the spatiotemporal sources of water taken up by plants (Miguez-Macho and Fan, 2021), calculating the mean transit time of various hydrological components (Dai et al., 2022), estimating the source water contribution to root water uptake (Dai et al., 2020), and analyzing the $WUE_i$ of a plant (Wu et al., 2022). Previous studies on plantation tree water-use characteristics have focused on comparing the same species in mixed and pure forests (Tang et al., 2018), different species within mixed plantations (Grossiord et al., 2014; Liu et al., 2021b; Tang et al., 2019), different types of pure forests (Wang et al., 2020), and different tree ages (Wang et al., 2021). Although comparisons of water-use strategies





between degraded trees (with altered organ structures and reduced functions) and healthy trees would be useful for clarifying

the mechanisms of tree degradation or mortality, such comparisons are still lacking.

Although seasonal patterns in deep soil water utilization help trees survive drought, they accelerate the exhaustion of deep soil reservoirs that are difficult to recharge and reduce the stability of plantations in water-limited areas (Chen et al., 2008; Su and Shangguan, 2019). Therefore, understanding the relative contributions of water sources and absolute water consumption of afforestation species is essential for assessing the soil water-carrying capacity for vegetation in water-limited

areas. Using stable isotope technology, coupling the thermal dissipation method to monitor sap flow, and then estimating the whole tree or stand transpiration (Granier, 1987), we can comprehensively grasp the relationship between trees and water, providing scientific evidence for forest management to prevent further declines.

This study was conducted on *Populus simonii* plantations in a semi-arid area of the Chinese Loess Plateau. *P. simonii* is the local vanguard tree for afforestation projects; however, in recent years, the structure of trees in these plantations has changed

and their function has declined in this area. First, we carried out a field investigation measuring tree growth indicators and stand structure parameters (height, size, canopy width, and dead branches) and then conducted a comprehensive evaluation to distinguish different degradation degrees of *P. simonii*. Then, the soil water content, hydrogen and oxygen isotopic compositions in the soil water and xylem water, carbon isotopic compositions in the leaf, and sap flow of trees were continuously measured at each sample site during the growing season. The objectives were to: (1) identify the isotopic

characteristics of soil water, xylem water, and leaves, (2) compare the water-use strategies (transpiration, root water uptake patterns, and $WUE_i$) under different categories of degradation, and (3) elucidate the interaction mechanism between tree degradation and soil water status. We hypothesized that with increased tree degradation, the deep roots and transpiration of *P. simonii* will decrease, which will lead to a reduction in the relative contribution and absolute use of deep soil water and thus a trend of increasing water storage in the deep layer.

## 2 Materials and Methods

### 2.1 Study area and sampling sites

This study was conducted at the Shenmu Erosion and Environment Experimental Station of the Chinese Academy of Sciences, which is located in the Liudaogou catchment of the Loess Plateau, China (110°21′E, 38°47′N) and has an area of 6.9 km$^2$ (Fig. 1a). The area is in an interlaced zone experiencing wind and water erosion, and it constitutes the transition

from the hilly loess area to Mu Us Desert. The catchment is a typical loess hilly landform with sand cover, and it belongs to a middle-temperate semiarid climate zone and had an average annual temperature of 8.4℃ and annual precipitation of 459 mm from 2003 to 2017. The precipitation from June to September accounts for 77.4% of the annual precipitation. Frequent sandstorms and intense wind erosion occur in spring, and rainstorms resulting in strong water erosion (i.e. interrill and gully erosion) occur mostly in summer.





The sampling sites were located on flat sandy land at the southern end of the Liudaogou catchment at an altitude of 1198 m and a slope of <2°. The soil texture is sandy (USDA classification), and the granular composition is 95.7% sand, 3.2% silt, and 1.1% clay. The field capacity and wilting coefficient of the soil are 0.06 and 0.01 $cm^3$ $cm^{-3}$, respectively. The vegetation is dominated by planted *P. simonii* trees (~800 tree $ha^{-1}$), with a planting area of ~150 000 $m^2$. The understory herbs are dominated by *Artemisia desertorum*, *Incarvillea sinensis,* and *Lespedeza bicolor*.

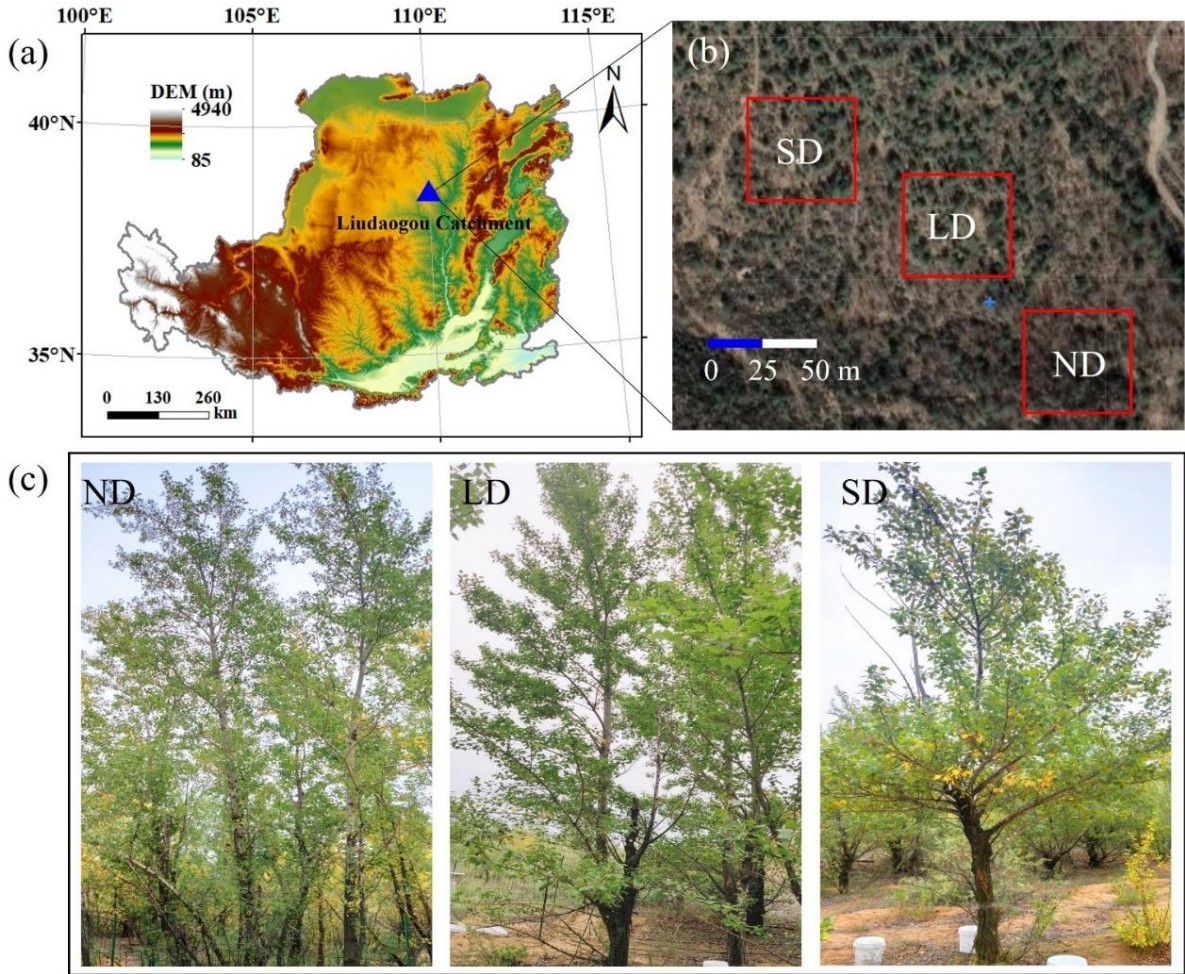


**Figure 1: Geographical location of this study area (a), location of sampling sites (b), and growth status of *P. simonii* (c). The satellite image in Fig. (b) is from https://www.earthol.com/g/, and the red areas represent fixed sampling sites. ND, LD, and SD indicate no, light, and severe degradation of *P. simonii*, respectively.**

### 2.2 Determination of sampling sites to reflect different categories of degradation

*P. simonii* plantations on the sandy land were surveyed to establish long-term sampling sites in early May 2021. Through a literature review and consultation with local farmers, we learned that *P. simonii* was planted in the study area around 1999.





The degradation degree of *P. simonii* was investigated in detail according to the growth status and spike top (Liu et al., 2020; Liu et al., 2021a). The "spike top" was described as a dead area of trunk in the upper canopy of *P. simonii* trees showing continuous growth of lateral branches (Fig. 1c). First, we determined three $50 \times 50$ m$^2$ plots with distinct growth differences in height, size, canopy width, and spike top, and their topographic factors were similar (Fig. 1b). *P. simonii* in the three plots experienced varying degrees of degradation after years of growth. Then, four $20 \times 20$ m$^2$ quadrats were set up in each plot, and the growth indicators of all *P. simonii* were measured, including the tree height, spike top height, diameter at breast height (DBH), and canopy projection area (A$_c$, calculated by the crown width in the east-west and north-south directions). The canopy density of each quadrat was obtained from the ratio of total A$_c$ to total area. We surveyed 128 trees in each of the sample plots. The basic information above is shown in Table 1. Notably, in one of the three plots, *P. simonii* had the best growth according to fine growth indicators and no dead branches in the upper trunk (Fig. 1c), and this plot was defined as no degradation (ND). The mean height, DBH, and A$_c$ of trees in the ND plot were 10.3 m, 19.2 cm, and 10.3 m$^2$, respectively. However, the growth status of *P. simonii* in the other two plots was inferior to that in the ND plot to varying degrees, and the *P. simonii* trees exhibited a spike top. These plots were classified as light degradation (LD) and severe degradation (SD) based on the growth indicators and spike top ratio (Liu et al., 2020; Liu et al., 2021a). The spike top ratio was derived by dividing the average spike top height by the average tree height. The trees in the LD and SD plots had a spike top ratio of 0–40% and >40%, respectively. *P. simonii* in the LD plot had a mean height, DBH, and A$_c$ of 9.3 m, 15.4 cm, and 9.0 m$^2$, respectively, while *P. simonii* in the SD plot had the worst growth parameters, with an average height, DBH, and A$_c$ of 5.9 m, 12.5 cm, and 5.9 m$^2$, respectively.

**Table 1  Basic characteristics of *P. simonii* at different sampling sites**

| Growth status | Tree height (m) | Spike top height (m) | Spike top ratio (%) | DBH (cm) | A$_c$ (m$^2$) | Crown density (%) |
|---|---|---|---|---|---|---|
| ND | $10.3 \pm 1.3$ | 0 | 0 | $19.2 \pm 3.8$ | $10.3 \pm 3.4$ | $83.4 \pm 13.2$ |
| LD | $9.3 \pm 1.0$ | $2.6 \pm 0.4$ | $28.0 \pm 3.5$ | $15.4 \pm 2.1$ | $9.0 \pm 1.9$ | $72.0 \pm 9.5$ |
| SD | $5.9 \pm 0.8$ | $2.7 \pm 0.3$ | $45.8 \pm 3.1$ | $12.5 \pm 1.6$ | $5.9 \pm 1.5$ | $47.2 \pm 8.4$ |

The DBH and A$_c$ represent diameter at breast height and canopy projection area, respectively. Values are shown as the mean ± standard deviation.

### 2.3 Sampling and isotope analysis

In May-September 2021, soil and plant samples were collected monthly from three fixed sites of $20 \times 20$ m$^2$ characterized as ND, MD, and SD. The specific sampling dates were May 24, June 28, July 28, August 27, and September 24. Since the variability in tree morphology appeared mainly among the sampling sites with different categories of degradation rather than within the same site (Table 1), we selected three sample trees at each fixed site as well as three soil profiles for sampling, and these sample trees had similar morphological indicators to the average of trees of each degradation class. For the xylem samples, three lignified twigs (~1.5 cm in diameter) at a height of 200 cm were cut off from the south-facing side of three sample trees. The twigs were peeled, sealed in a 12-mL glass bottle with parafilm, and brought to the laboratory for frozen storage. Three replicate leaf samples were collected from the ND, LD, and SD trees. Mature and healthy leaves in the upper



layer of the canopy under no shading conditions were cut off. Fresh leaves (~40 g) were then packed in paper bags and brought to the laboratory for pretreatment, which included drying at 75°C for 48 h in an electric heating air-blowing drier, grinding to pass through an 80-mesh screen, and sealing in a polyethylene bag for preservation at 15°C. We collected a total

of 45 xylem samples and 45 leaf samples during the observation period.

Soil samples were obtained using a hand drill (40 mm in diameter) at a distance of 1.5 m from the sample trees (three replicates per degradation class). The sampling depth was 0–200 cm, and the sampling intervals were 10 cm and 20 cm at the depth of 0–20 cm and below 20 cm, respectively. We collected a total of 495 soil samples during the observation period. The samples were divided into two parts. One part was oven dried to measure the soil gravimetric water content, and the other

part was sealed into a 12-mL glass bottle with parafilm and brought to the laboratory for frozen storage. A cryogenic vacuum distillation system (Li-2100, LICA Inc., Beijing, China) was employed to extract water from the xylem and soil, and the water extraction rate was over 99%. For the specific operational processes, see Zhao et al. (2021).

In August 2021, a root investigation of *P. simonii* was carried out using a root drill (90 mm in diameter) in the 0–200 cm soil profile. One sample tree trunk was selected as the origin in the ND, LD, and SD plots, and no shrubs and few herbs were

located nearby, and three soil cores at a depth of 0–200 cm were taken from 50, 100, and 200 cm away from the trunk in a southerly direction. The sampling interval was the same as that used for the soil samples. Each soil layer was passed through a screen (2 mm in diameter) for root-soil separation. Fine roots (<2 mm in diameter) were sealed in polyethylene bags for preservation, and the root weight density was determined using the oven-drying method.

Precipitation samples were obtained from May to September in 2021 (*N*=15) using a rain gauge placed on the open ground

~1000 m away from the *P. simonii* sampling sites. Because light rainfall events (precipitation <3 mm) only moistened a few centimeters of soil in the *P. simonii* plantation, we collected precipitation samples in 30-mL polyethylene bottles after each rainfall event with precipitation >3 mm, immediately sealed them with parafilm, and transported them to the laboratory for frozen storage.

The hydrogen and oxygen isotopic compositions of the precipitation and soil water were analyzed using a liquid-water

isotopic analyzer (912-0032, LGR Inc., California, USA). The hydrogen and oxygen isotopic compositions of xylem water and the carbon isotopic compositions of leaf samples were analyzed using a stable isotope ratio mass spectrometer (Isoprime-100, Isoprime Inc., Cheadle, UK). Isotopic compositions are expressed using delta notation as follows:

$$\delta X(‰) = R_{sample}/R_{standard} - 1 , \tag{1}$$

where $\delta X$ represents the $\delta^2H$, $\delta^{18}O$, and $\delta^{13}C$, and $R_{sample}$ and $R_{standard}$ are the isotopic ratios of the measured and standard

samples ($^2H/^1H$, $^{18}O/^{16}O$, and $^{13}C/^{12}C$), respectively. The standard samples were Vienna Standard Mean Ocean Water (VSMOW) for $\delta^2H$ and $\delta^{18}O$ and Pee Dee Belemnite (PDB) for $\delta^{13}C$. The accuracies of $\delta^2H$, $\delta^{18}O$, and $\delta^{13}C$ were 0.5‰, 0.1‰, and 0.1‰, respectively.



## 2.4 Soil water content

At each sample site (ND, LD, and SD), seven soil moisture sensors (10HS, METER Group, Inc., Washington, USA) were
installed in a 0–200 cm soil profile at a distance of ~2 m from the sample tree to continuously monitor the soil volumetric
water content (SWC) at various depths (10, 20, 40, 80, 120, 160, and 200 cm) from June to September in 2021. The
sampling interval was 30 min. The SWC monitoring data were calibrated using the oven drying method and cutting ring
method. The calibration process was described in Dai et al. (2023). The time series of the SWC at each depth monitored by
sensors and manual measurement are shown in Fig. A1 and Fig. A2 of the Appendix A, respectively, and their correlation
coefficient is 0.87 ($P<0.001$, $N=210$). Because our experimental plots were located on flat sandy areas and the soil texture is
homogeneous (the soil sand content is >95%), the spatial variability of the SWC within the same plot was considered
insignificant, and the continuous SWC monitoring data of one soil profile in each plot can reflect the temporal variation of
SWC in the plot. Moreover, the differences in SWC between different experimental plots were assumed to be related to
vegetation water consumption.

Relative extractable soil water (REW) was used to evaluate the dry and wet conditions of soil using the following formula
(Iqbal et al., 2021):

$$\text{REW} = \left( \frac{\text{SWC}_{0-200} - \text{SWC}_{\min}}{\text{SWC}_{\max} - \text{SWC}_{\min}} \right), \tag{2}$$

where $\text{SWC}_{0-200}$ is the average SWC ($\text{cm}^3 \text{ cm}^{-3}$) at 0–200 cm depth and $\text{SWC}_{\min}$ and $\text{SWC}_{\max}$ are the minimum and
maximum average SWC at 0–200 cm depth during the growing season, respectively. A REW less than 0.4 indicates that the
soil is in the dry stage, and a REW $\geq 0.4$ indicates that the soil is in the wet stage (Zhou et al., 2013).

## 2.5 Sap flow measurements

At each sample site (ND, LD, and SD), the sap flow velocity of five trees close to the soil and plant sampling sites was
continuously monitored using a thermal dissipation probe (TDP-20, Dynamax Inc., TX, USA) at 1.3 m above the ground on
the north side from June to September in 2021 (*P. simonii* leaves fall extensively at the beginning of October). Data were
recorded by loggers (CR-1000, Campbell Scientific, Utah, USA) in 30 min intervals. The height, DBH, and $A_c$ of these
sample trees were close to the mean of each sample site. The TDP-20 consisted of upper and lower probes (20 mm in length
and 2 mm in diameter) containing a copper-constantan thermocouple, and the probes were used for heating and as a
reference, respectively. The temperature difference ($\Delta T$, ℃) between the two probes and the maximum $\Delta T$ ($\Delta T_m$, ℃) (when
the sap flow was near zero) were calculated, and the sap flow velocity ($F_d$, $\text{cm}^3 \text{ cm}^{-2} \text{ h}^{-1}$) was calculated according to the
empirical calibration formula proposed by Granier (1987):

$$F_d = 0.0119 \left( \frac{\Delta T_{\max} - \Delta T}{\Delta T} \right)^{1.231} \times 3600, \tag{3}$$

The sapwood area ($A_s$, $\text{cm}^2$) thickness was derived from sapwood thickness investigations of the sample trees using an
increment borer and a vernier caliper at the end of the growing season. The mean sapwood areas of the sample trees from the





ND, LD, and SD plots were 220.9, 114.4, and 95.9 cm$^2$, respectively. The daily transpiration (Tr, mm d$^{-1}$) of the whole tree

scale was calculated as follows:

$$Tr = (F_d \times A_s \times 24)/(A_c \times 10^3) ,$$                    (4)

The Tr of each degradation class was the average value of five sample trees in the ND, LD, and SD plots, and the time series

of Tr of each sample tree are shown in Fig. A3 of the Appendix A.

**2.6 Statistical analysis**

In this study, soil water at various depths was considered the water source for *P. simonii*. Because the groundwater level in

the study area exceeded 40 m (Chen et al., 2008), the roots could not absorb and utilize groundwater; moreover, vegetation

was not irrigated at the sampling sites. According to the SWC and soil water isotope values, three soil layers in the 0–200 cm

profile with significant differences ($P<0.05$) in isotopic compositions were selected as potential water sources: shallow (0–40

cm), middle (40–80 cm), and deep (80–200 cm) layers. The shallow layer had the most variability in soil water isotopes and

SWC and was most susceptible to rainfall pulses and evaporation. The middle layer had relatively large variability in soil

water isotopes and SWC and was vulnerable to heavy rainfall events. The deep layer had relatively stable variations in soil

water isotopes and SWC and was poorly influenced by precipitation and evaporation. The relative contribution of each

potential water source to the root water uptake of *P. simonii* was calculated using the MixSIR model with a Bayesian

framework (Moore and Semments, 2008). The MixSIR model can analyze multiple sources of water and incorporate the

uncertainties of isotopic compositions and their fractionations in the establishment of posterior probability distributions of

source water contributions. We assumed no isotopic fractionation during root water uptake (Dawson and Ehleringer, 1991)

and set the fractionation coefficient to zero. We imported the source and mixture data ($\delta^2$H and $\delta^{18}$O in xylem water) into the

model using MATLAB 8.0 (MathWorks Inc., Natick, USA) and then set the iterations to one million times. The average

relative contributions of each source to the posterior probability distributions were analyzed.

The differences in root weight density, isotopic compositions ($\delta^2$H, $\delta^{18}$O, and $\delta^{13}$C) of soil and plant, daily transpiration, and

the proportion of root water uptake in the ND, LD, and SD plots were analyzed using one-way ANOVA, and the Tukey LSD

method was used for post hoc multiple comparisons of variables. Statistical analyses were performed using SPSS 22 (IBM

Inc., Chicago, USA).

**3. Results**

**3.1 Horizontal and vertical root distributions**

The horizontal root system of *P. simonii* was well-developed, and the average root weight densities at 200 cm from the trunk

were 483.7, 399.9, and 235.1 g m$^{-3}$ in the 0–200 cm soil profile in the ND, LD, and SD plots, respectively. No significant

difference ($P>0.05$) in the average root weight density was found among the different distances from the trunk (50, 100, and



200 cm) in the ND, LD, and SD plots (Fig. 2). In the vertical root system, the root weight density decreased with increasing soil depth. The average root weight density in the 0–200 cm soil profile was 741.4, 559.8, and 446.3 g m$^{-3}$ in the ND, LD, and SD plots. The differences between these plots were significant ($P<0.05$). As degradation intensified, the root proportion of the shallow layer gradually increased while that of the middle and deep layers decreased. The root proportions of the shallow, middle, and deep layers were 70.1%, 9.3%, and 20.6% in the ND plot; 81.0%, 8.3%, and 10.6% in the LD plot; and 91.2%, 6.0%, and 2.8% in the SD plot, respectively.

**Figure 2: Horizontal and vertical root distributions of *P. simonii*. The box plots show the minimum, 25% percent, median, 75% percent, and maximum of root weight density at 50, 100, and 200 cm from the trunk in the 0–200 cm soil profile, and black squares indicate the mean. The inset shows a properly enlarged vertical root distribution on the X-axis.**





## 3.2 Isotopic compositions of different water pools

The cumulative precipitation was 205.4 mm from May to September, with September having the lowest precipitation at 15.5 mm (Fig. 3a). The SWC of each potential water source (shallow, middle, and deep layers) for *P. simonii* is shown in Fig. 3b-d. The SWC of the shallow layer ranged from 0.019 to 0.066 $cm^3$ $cm^{-3}$ and responded rapidly to rainfall events. No significant difference ($P>0.05$) in the mean SWC of the shallow layer was found among the ND, LD, and SD plots (Fig. 3b). The SWCs of the middle and deep layers were not sensitive to rainfall events and showed a decreasing trend from May to September. We found significant differences ($P<0.05$) in the average SWCs of the middle and deep layers among the ND, LD, and SD plots (Fig. 3c, d). We also found that ND had the lowest average SWCs in the middle and deep layers while LD and SD had the highest average SWCs in the middle and deep layers, respectively. These results indicated that *P. simonii* forests without degradation may require more soil moisture from the middle and deep layers than forests with light and severe degradation.

The $\delta^2H$ and $\delta^{18}O$ in precipitation varied from –151.78 to –6.99‰ and from –20.52 to –0.40‰ (Fig. 3a), and the precipitation-weighted averages (± standard deviation) were –48.65 ± 37.60‰ and –7.72 ± 5.17‰, respectively. The maximum and minimum monthly mean $\delta^2H$ and $\delta^{18}O$ in precipitation occurred in June and September, respectively. In the dual-isotope space (Fig. 4), the slope and intercept of the local meteoric water line (LMWL, $\delta^2H=7.17\delta^{18}O+3.19$, $R^2=0.97$) were smaller than those of the global meteoric water line (GMWL, $\delta^2H=8\delta^{18}O+10$) (Craig, 1961), thus reflecting the arid climate of the study area. With increasing depth, the scatter points of $\delta^2H$-$\delta^{18}O$ in the soil water gradually approached and clustered toward the LMWL. The relationships of $\delta^2H$-$\delta^{18}O$ in the soil water (soil water evaporation line, SWL) in the ND, LD, and SD plots showed similar characteristics (ND: $\delta^2H=4.29\delta^{18}O–33.21$, $R^2=0.81$; LD: $\delta^2H=4.29\delta^{18}O–34.59$, $R^2=0.87$; SD: $\delta^2H=4.05\delta^{18}O–35.77$, $R^2=0.82$). The scatter points of $\delta^2H$-$\delta^{18}O$ in the xylem water matched those in the soil water in the dual-isotope space (Fig. 4), thus implying a lack of apparent deuterium depletion. We observed no significant differences ($P>0.05$) in the mean $\delta^2H$ and $\delta^{18}O$ in the soil and xylem waters among the ND, LD, and SD plots (Table 2). Although seasonal variations in $\delta^2H$ and $\delta^{18}O$ in the shallow soil water were similar to those in precipitation (i.e., the maximum in June and the minimum in September), such distinct seasonal variations in $\delta^2H$ and $\delta^{18}O$ were not observed in the middle and deep soil water in the ND, LD, and SD plots (Fig. 3 and Fig. 5). With increasing depth, soil water isotopes were gradually depleted in the ND, LD, and SD plots (Fig. 5 and Table 2).



**Figure 3: Temporal variations in daily precipitation (Pr), δ²H-δ¹⁸O in precipitation (a), and SWC at various depths (b–d). The SWC in May is obtained from the soil gravimetric water content and soil bulk density. The box plots show the minimum, 25% percent, median, 75% percent, and maximum of soil water content. Different lowercases in the**



**270** **box plots denote a significant difference in SWC of the same soil layer among the sample sites ($P<0.05$). The orange arrows correspond to the sampling dates of soil and plant isotopes.**



**Figure 4: Scatter distributions of $\delta^2H$ and $\delta^{18}O$ in precipitation, soil water, and xylem water. LMWL represents the local meteoric water line ($\delta^2H=7.17\delta^{18}O+3.19$, $R^2=0.97$), and GMWL represents the global meteoric water line ($\delta^2H=8\delta^{18}O+10$). SW$_{0–40}$, SW$_{40–80}$, and SW$_{80–200}$ represent the soil water in shallow (0–40 cm), middle (40–80 cm), and deep (80–200 cm) layers, respectively.**









**Figure 5: Seasonal variations in $\delta^2$H (a) and $\delta^{18}$O (b) in the soil and xylem water. Values are shown as the mean ±**
**standard deviation.**

**Table 2 Mean $\delta^2$H and $\delta^{18}$O in the soil and xylem water of *P. simonii* during the growth season**

| Sampling site | | Shallow soil water | Middle soil water | Deep soil water | Xylem water |
|---|---|---|---|---|---|
| $\delta^2$H | ND | −48.71 ± 17.32 | −64.09 ± 4.94 | −73.88 ± 3.12 | −58.43 ± 12.00 |
| (‰) | LD | −50.11 ± 18.43 | −65.53 ± 6.89 | −75.90 ± 4.14 | −61.85 ± 13.02 |
| | SD | −52.54 ± 18.43 | −59.65 ± 8.15 | −72.32 ± 3.16 | −59.97 ± 11.75 |
| $\delta^{18}$O | ND | −3.89 ± 3.62 | −7.53 ± 0.82 | −9.18 ± 0.62 | −7.28 ± 1.11 |
| (‰) | LD | −3.76 ± 4.11 | −7.66 ± 0.94 | −9.45 ± 0.65 | −7.70 ± 1.10 |
| | SD | −4.05 ± 3.64 | −6.86 ± 1.03 | −8.85 ± 0.49 | −7.07 ± 0.85 |

Values are shown as the mean ± standard deviation.

## 3.3 Characteristics of transpiration

The Tr in the ND, LD, and SD plots ranged from 0.04 to 2.38 mm d$^{-1}$, 0.03 to 1.30 mm d$^{-1}$, and 0.06 to 1.50 mm d$^{-1}$,
respectively (Fig. 6). The average Tr in the ND, LD, and SD plots were 0.93 ± 0.48, 0.65 ± 0.27, and 0.53 ± 0.26 mm d$^{-1}$,
respectively, and the difference among them was significant ($P<0.05$), indicating that tree degradation reduced water
consumption. From June to September, an overall trend of reducing REW was observed for each site (Fig. 6b–d), which
corresponded to the reduction in SWC for different soil layers (Fig. 3b–d), thus reflecting a decrease in soil available water
by trees. The Tr showed similar seasonal variation patterns among the experimental sites and all varied synchronously with
REW, e.g., Tr increased when REW increased (Fig. 6b–d). The results indicated that transpiration of *P. simonii* in this area
was closely related to the soil water conditions. The average Tr in the ND, LD, and SD plots during the soil wet period
(REW≥0.4) were 1.34 ± 0.50, 0.96 ± 0.26, and 0.77 ± 0.27 mm d$^{-1}$, respectively, which were significantly ($P<0.05$) higher
than that during the soil dry period (REW<0.4), with the values of 0.75 ± 0.48, 0.56 ± 0.27, and 0.43 ± 0.26 mm d$^{-1}$,
respectively.



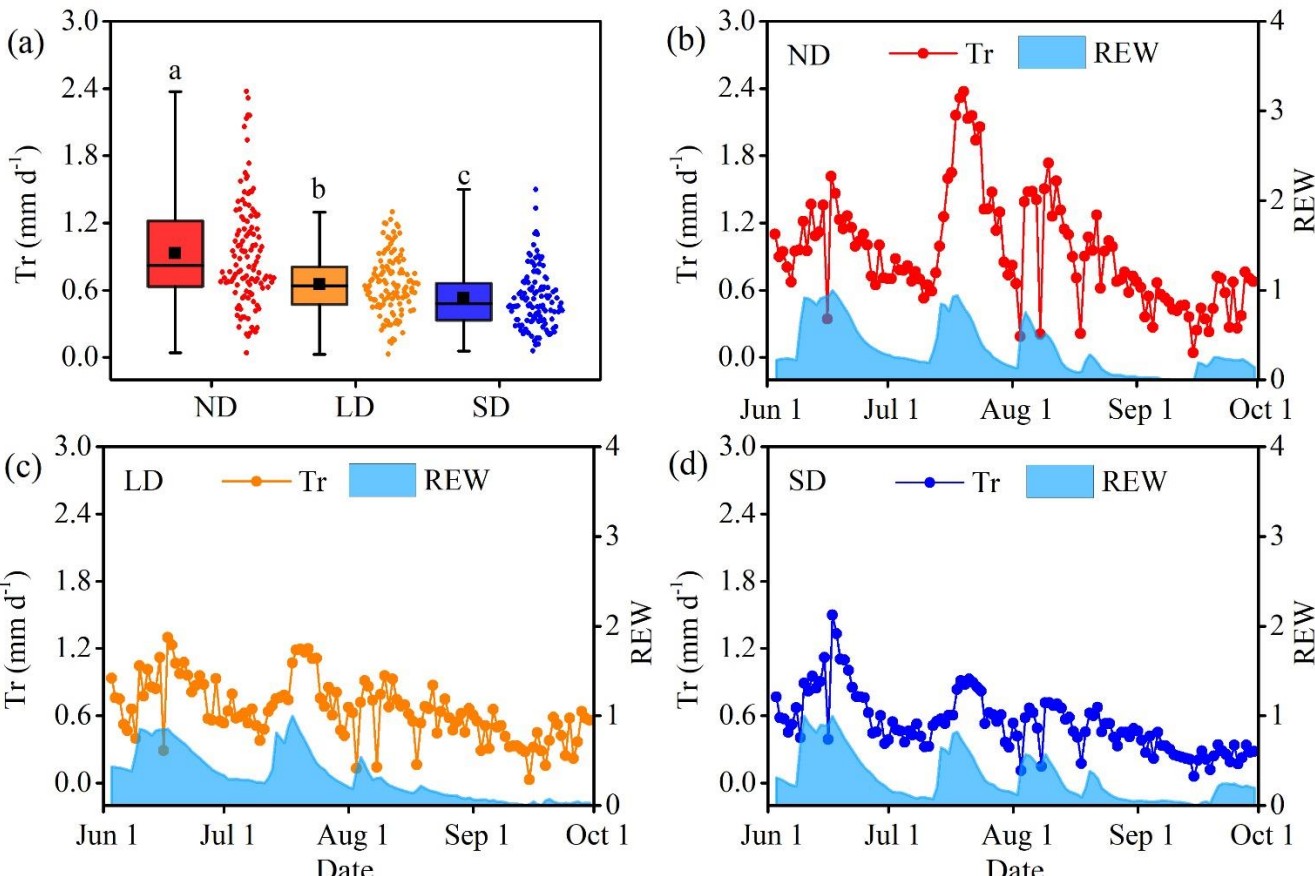

**Figure 6: Average transpiration of degraded *P. simonii* from June to September (a), Seasonal variation in transpiration and REW (b–d). Tr and REW represent transpiration and relative extractable water, respectively. The box plots show the minimum, 25% percent, median, 75% percent, and maximum of transpiration, and black squares indicate the mean. Different lowercases in the box plots denote a significant difference in Tr among the sample sites (*P*<0.05).**

**3.4 Water source apportionment**

The water absorption depths and proportions of *P. simonii* in the ND, LD, and SD plots were identified according to the MixSIR model. Similar seasonal variations in the main water absorption depths of *P. simonii* were observed in the ND, LD, and SD plots. *P. simonii* in the ND and LD plots mainly absorbed shallow soil water in May (the month when the soil was the wettest), with contributions of 63.1% and 61.7%, respectively, while *P. simonii* in the SD plot mainly absorbed middle soil water (48.0%) in May (Fig. 7a, b). *P. simonii* in the ND, LD, and SD plots mainly absorbed middle soil water from June to August, with contribution proportions of 42.1–45.0%, 38.3–45.2%, and 42.4–47.7%, respectively (Fig. 7a). *P. simonii* in the ND, LD, and SD plots mainly absorbed deep soil water in September (the month when the soil was the driest), with



proportions of 45.9%, 51.6%, and 49.8%, respectively (Fig. 7a, b). The results showed that *P. simonii* in the ND, LD, and SD plots can shift its water source from shallow to deep layers in the process of soil wetting to drying.








**Figure 7: Seasonal variations in the contribution proportions of water sources (a), mean REW (relative extractable water) (b), and absolute consumption of water sources (c) to _P. simonii_ water absorption. The absolute water consumption is the product of source water contribution proportions and monthly cumulative transpiration of _P. simonii_. Values are shown as the mean ± standard deviation.**

Based on the daily transpiration data of each site shown in Fig. 6, the absolute monthly consumption of soil water was calculated. From June to September, _P. simonii_ in the ND, LD, and SD plots consumed 14.3–38.1, 12.2–23.2, and 8.1–21.6 mm of soil water storage per month, respectively (Fig. 7c). Although the relative contribution proportion of deep soil water absorbed by _P. simonii_ was the highest in September, the absolute use of the water source in this month was even lower than that in June or July (Fig. 7c). In other words, the high relative contribution proportion of deep soil water during severe soil

drought conditions did not increase the absolute water use. Meanwhile, severe soil drought decreased the water use of the shallow and middle soil waters, and the combined effects resulted in extremely low transpiration in September.

### 3.5 Leaf $\delta^{13}$C and WUE$_i$

The leaf $\delta^{13}$C values of _P. simonii_ in the ND, LD, and SD plots ranged from –27.54 ‰ to –26.93‰, –27.56 ‰ to –26.59‰, and –27.17 ‰ to –26.59‰, respectively (Fig. 8a), and thus did not show significant seasonal variation ($P>0.05$). The mean

leaf $\delta^{13}$C values in the ND, LD, and SD plots were –27.2 ± 0.2‰, –27.1 ± 0.4‰, and –27.0 ± 0.2‰, respectively, and significant differences were not found among them ($P>0.05$). A weak positive correlation ($P>0.05$) was observed between leaf $\delta^{13}$C and SWC in the ND plot, whereas significant negative correlations ($P<0.05$) were observed between leaf $\delta^{13}$C and SWC in the LD and SD plots (Fig. 8b). The leaf $\delta^{13}$C of C$_3$ plants is generally associated with the WUE$_i$, and a larger leaf $\delta^{13}$C value corresponds to a higher WUE$_i$ (Wang et al., 2020; Zhao and Wang, 2021). The results showed that although tree

degradation did not directly affect WUE$_i$, it changed the sensitivity of WUE$_i$ to soil water.

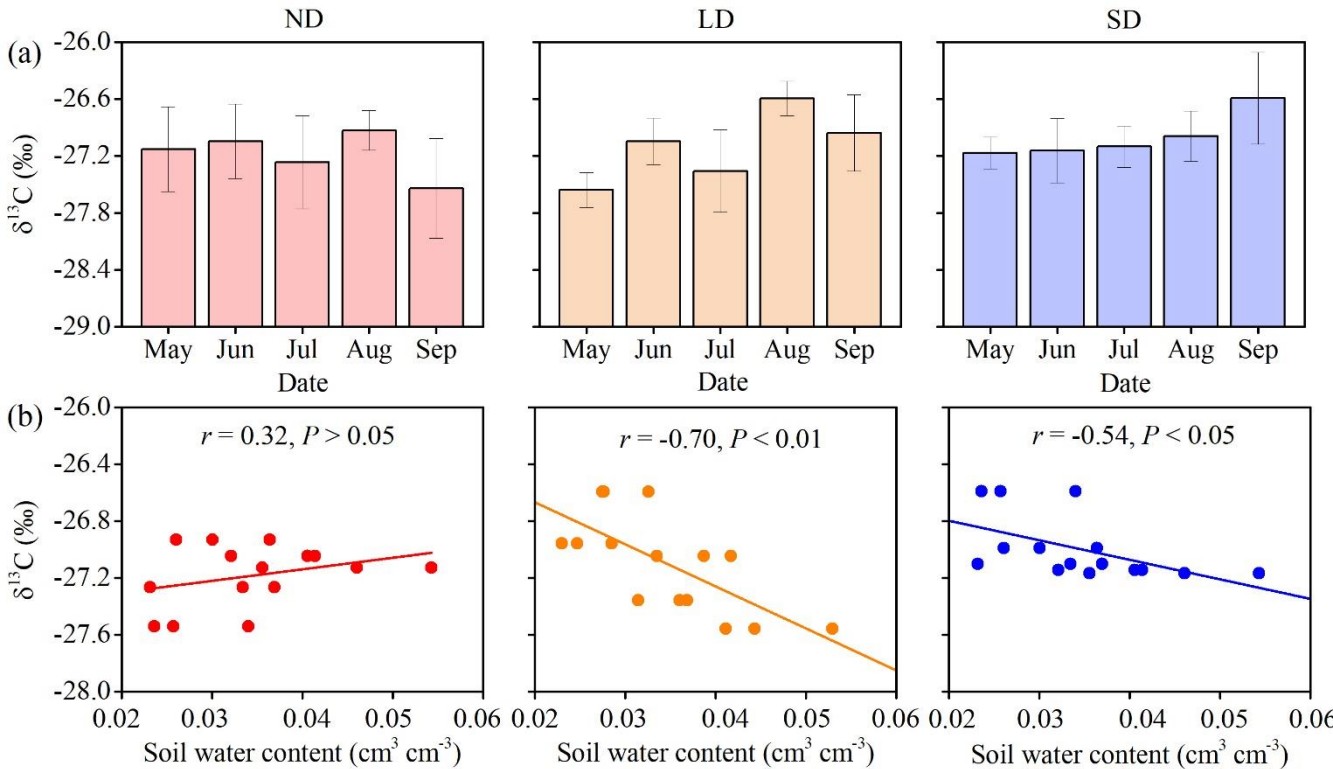

Figure 8: Seasonal variations in the average leaf δ¹³C (± standard deviation) of *P. simonii* (a) and their relationship with soil water content (b). Soil water content is the monthly average SWCs in the shallow (0–40 cm), middle (40–80 cm), and deep (80–200 cm) layers.

## 4 Discussion

### 4.1 Isotopic compositions of different water pools

In our study, the soil and xylem water isotopes did not significantly differ ($P>0.05$) among the ND, LD, and SD plots (Fig. 4, Fig. 5, and Table 2). Studies have shown that soil water isotopes are typically influenced by precipitation (Robertson and Gazis, 2006), evaporation (Lyu et al., 2021), and infiltration (Yang and Fu, 2017). Shallow soil water acquires most of the isotopic signals during precipitation and presents similar seasonal isotopic variations as precipitation. The slopes of the SWLs in the ND, LD, and SD plots were similar, reflecting the similar characteristics of soil evaporation at various sites (Lyu et al., 2021). Yang and Fu (2017) showed that when only one obvious peak soil water isotope value (close to that of recent precipitation) appeared in the vertical profile, then precipitation infiltrated unsaturated soil in the form of piston flow. However, when more than one peak soil water isotope value appeared in the vertical profile, then an increase in SWC occurred that may indicate preferential flow. Based on the above method, precipitation infiltration occurred in the form of piston flow in the ND, LD, and SD plots (Fig. 3 and Fig. 5). In summary, we inferred that similar precipitation infiltration



and hydraulic redistribution in the ND, LD, and SD plots might result in the stability of soil water isotopes. However, Liu et al. (2020) reported differences in soil water isotopes in *P. simonii* plantations under different degrees of degradation, which could be attributed to changes in the soil physical properties and groundwater recharge.

Although *P. simonii* without degradation showed a relatively abundant and deep root system (Fig. 2) and may prefer to utilize more deep soil water (relatively depleted in heavy isotopes), resulting in more depleted isotopes in its xylem water, the low soil water availability in the deep layer (Fig. 3) limited the water uptake from deep soils. However, degraded *P. simonii* had a relatively sparse and shallow root system (Fig. 2) and were able to absorb a certain amount of water from the deep layer due to high soil water availability (Fig. 3). Therefore, the water source apportionment of *P. simonii* may have

been consistent across the different sites and the water isotopes in the plant xylem did not necessarily differ in the ND, LD, and SD plots. The xylem water isotopes of *P. simonii* were consistent with the soil water isotopes (Fig. 4), indicating that no obvious isotopic fractionation occurred during root water uptake and water transport, which is a prerequisite for determining the water source apportionment based on the stable isotope method. This result was inconsistent with that obtained by Zhao and Wang (2021) for *Salix matsudana* at a check-dammed channel in the catchment, which this discrepancy might be related

to the soil water conditions (Zhao et al., 2022).

**4.2 Water-use strategy of *Populus simonii***

As degradation intensified, tree transpiration significantly decreased (*P*<0.05). The higher the degradation degree, the lower the DBH, crown width, and root biomass of the tree (Fig. 1 and Fig. 2) and the less water needed for tree growth. According to the MixSIR model, *P. simonii* shifted its water source from shallow to deep layers in the process of soil wetting to drying

(Fig. 7a, b). The same results were found for *Pinus tabuliformis* and *Hippophae rhamnoides* in semi-arid regions (Tang et al., 2018). Moreover, the root water uptake patterns were similar in terms of both time and depth in the ND, LD, and SD plots (Fig. 7a), indicating that the ability of degraded trees to absorb deep soil water had not changed. The findings were inconsistent with those of Liu et al. (2020) and Liu et al. (2021a). In these studies, as degradation intensified, *P. simonii* gradually lost its ability to obtain water from deep soil, which was related to the smaller tree size and shallower root

distribution. However, in our study, the undiminished plasticity in deep soil water absorption of severely degraded *P. simonii* might be associated with the fact that a small number of roots were still distributed in the deep layer (Fig. 2c). Furthermore, a potential water source for tree utilization was provided by the deep layer because of the higher SWC (Fig. 3d).

Surprisingly, during the severe water stress in September, deep soil contributed the largest proportion of water for transpiration in *P. simonii* but did not compensate for the low water availability of the shallow and middle layers (Fig. 7),

which was consistent with the findings of Gessler et al. (2022). These researchers found that beech did not have a compensating mechanism for deep roots during water stress and that water use depended mainly on water availability in the upper soil. We accepted the explanation that deep roots become impotent because soil drought severely inhibits underground metabolism, carbon allocation to roots, and root growth (Joseph et al., 2020). In addition, this effect might also be related to the depletion of available water in the deep layer. In the late growing season, the nearly exhausted deep soil water



compensates little for the shortage of shallow soil water (Fig. 3). Therefore, the increased relative contribution proportion of deep water absorbed by trees during drought should be considered along with the absolute consumption of deep water by trees because trees might suffer degradation or mortality when deep soil water is insufficient to support normal growth. Zhao and Wang (2021) suggested that leaf $\delta^{13}C$ is related to photorespiration, carbon isotopic fractionation after photosynthesis, and changes in environmental conditions. The variations in environmental conditions (temperature, humidity,

and atmospheric carbon dioxide concentration) in the ND, LD, and SD plots were the same, as was the variation in carbon assimilation and transpiration in the same species, which might have resulted in the consistency among leaf $\delta^{13}C$ and $WUE_i$ (Fig. 8a). Interestingly, the relationships between leaf $\delta^{13}C$ of *P. simonii* and SWC were not consistent in the ND, LD, or SD plots (Fig. 8b). Leaf $\delta^{13}C$ in the ND plot showed a weak positive correlation with SWC, which is similar to the results of Wang et al. (2020). In contrast, leaf $\delta^{13}C$ in the LD and SD plots showed a negatively correlated with SWC ($P<0.05$), which

was consistent with the results of Ale et al. (2018). With decreasing soil water availability, reduced stomatal conductance of *P. simonii* often resulted in a lower ratio of intercellular to atmospheric partial pressure of $CO_2$, and then the leaf $\delta^{13}C$ and $WUE_i$ increased (Prentice et al., 2011). These findings implied that deteriorating *P. simonii* trees could enhance their drought adaptability by improving $WUE_i$ under soil water stress.

### 4.3 Interaction mechanism between tree degradation and soil water status

*P. simonii* is a fast-growing tree with high canopy transpiration demands (Wang et al., 2019) and survival ability (Sun et al., 2018), and in the 1990s, it represented the local vanguard tree species for afforestation and restoration in this study area. However, in recent years, deep soil desiccation has been observed, and vegetation afforestation in semi-arid areas has been implicated (Zhang et al., 2018). When the water source in the upper soil is insufficient to meet the demand for tree growth, the water use in the deep layer increases, and with long-term insufficient amounts of precipitation in water-limited areas, soil

desiccation occurs. Meanwhile, increasing studies have also reported degradation of these afforestation species (Chen et al., 2015; Liu et al., 2020; Sun et al., 2018). The link between tree degradation and soil water status has received increasing attention (Liang et al., 2022; Liu et al., 2020). In this study, due to the large tree transpiration and developed root system in the ND plot, the SWCs in the middle and deep layers were significantly lower than those in the LD and SD plots during the growing period (Fig. 2, Fig. 3, and Fig. 6). Moreover, precipitation cannot easily infiltrate into the middle and even deep

layers in this study area (Fig. 3), which hinders the replenishment of deep water reserves. In the long run, soil water conditions undoubtedly hindered the growth of *P. simonii*, which has high water demands. According to McDowell et al. (2008), under prolonged water stress, trees might suffer from carbon starvation and a cascade of downstream effects caused by the avoidance of drought-induced hydraulic failure via stomatal closure. In addition, *P. simonii* without degradation showed greater height, size, and canopy and tended to carry a higher risk of hydraulic failure than *P. simonii* with light and

severe degradation (Bennett et al., 2015). Therefore, tree degradation might occur in normally growing *P. simonii* (from ND to LD) in the long-term under low soil water availability. On the contrary, *P. simonii* in the SD plot prevented water depletion by reducing the root biomass and transpiration, which improved the deep soil water status to some extent (Fig. 2,



Fig. 3, and Fig. 6). Thus, tree degradation in the SD plot might be reduced (from SD to LD) in the long-term, which reflects the tenacity of *P. simonii* in harsh environments.

In summary, water shortage is the main problem faced by *P. simonii* plantations in the study area. To maintain healthy growth of *P. simonii* and fully exploit the ecological functions of existing *P. simonii* plantations, forest management to modify stand structure should be carried out. To be specific, we should consider the bearing capacity of vegetation in relation to soil moisture and forest thinning to ensure that precipitation matches tree growth and gradually replenishes the deep soil reservoir. For *P. simonii* plantations without degradation, we can remove some of the trees, trim the lower canopy

side branches to reduce the canopy density, and cover the ground with these branches, to increase net precipitation and reduce soil evaporation. For *P. simonii* plantations showing degradation, which also have played an important role in recent years in wind prevention and sand fixation, the main management measure is to cover the ground with pruned branches and leaves from *P. simonii* without degradation to reduce the higher soil evaporation caused by the low canopy density, thereby accelerating the recovery of the soil water reservoir. Additionally, a mixed tree-shrub pattern should also be considered to

increase biodiversity. Hydrological niche separation between species in mixed forests can improve resource allocation (Moreno-Gutiérrez et al., 2012) and alleviate water depletion in deep soils to a certain extent (Wang et al., 2020).

**5 Conclusion**

In this study, we found no obvious difference in the isotopic characteristics of the soil water and xylem water among the ND, LD, and SD plots, which showed similar seasonal isotopic patterns, SWLs slopes, and isotopic profile distributions. As tree

degradation intensified, the root biomass decreased significantly ($P<0.05$) and the root proportion of the deep layer decreased. Influenced by precipitation, significant differences were not observed in the SWC ($P>0.05$) of the shallow layer among the ND, LD, and SD plots, although the SWCs of the middle and deep layers in the ND plot were significantly lower than those in the LD and SD plots ($P<0.05$). Compared with trees without degradation, degraded trees showed significantly reduced transpiration ($P<0.05$) but did not show changes in root water uptake patterns. *P. simonii* under different degradation

classes shifted the water source from shallow to deep layers in the process of soil wetting to drying. Thus, the original hypothesis regarding the root water use patterns was rejected, while the other hypotheses were supported. Additionally, the sensitivity of $WUE_i$ to SWC for degraded trees increased compared to that of the trees without degradation. In conclusion, although degraded trees alleviated water depletion in deep soils to a certain extent, large-scale arbor planting causes soil desiccation due to the larger transpiration demand. The low soil water availability will inhibit tree growth, which is not

conducive to their ecological function. Therefore, stand management that involves proper thinning and a mixed tree-shrub pattern may be necessary to improve the soil water status.



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





## Appendix A



Fig. A1: Daily variation in soil volumetric water content (SWC) at different depths for each degradation class
595   monitored by sensors from June to September. ND, LD, and SD indicate no, light, and severe degradation of *P. simonii*, respectively.





**Fig. A2: Daily variation in soil volumetric water content (SWC) at different depths for each degradation class monitored by manual measurement (soil gravimetric water content times soil bulk density) from June to September.**
600    **ND, LD, and SD indicate no, light, and severe degradation of *P. simonii*, respectively.**





**Fig. A3: Daily variation in transpiration in five sample trees of each degradation class from June to September. ND, LD, and SD indicate no, light, and severe degradation of *P. simonii*, respectively.**





**Data availability**

605 The data that support the findings of this study are available from the corresponding author upon request.

**Author contributions**

JD and LW conceptualized this research. YZ collected the data. All authors contributed to the writing of the manuscript.

**Competing interests**

The authors declare that they have no conflict of interest.