# Peer review of "Reduced transpiration without changes in root water uptake patterns in degraded trees in semi-arid afforestation ecosystems"

_Hydrology and Earth System Sciences, 2023_

## Referee Comment (RC1)

HESSd: Reduced transpiration without changes in root water uptake patterns in degraded trees in semi-arid afforestation ecosystems

Title: Not exactly precise. The study wants to inform about afforestation systems about succeptability to drought. The title is too specific and also if that is the singular result it is not very novel or surprising.

Abstract:

Stable isotope mthod: what is that supposed to mean? Sloppy formulation

Hydrologic redistribution: How do you know? This is an interpretation based on what data?

General comment: The results are summarized in a very based and interpreted way and the concluding sentence does not really fit the described results. Also the different methods seem to contradict each other (e.g. different indication of root water uptake pattern from soil moisture compared to isotopic approaches)

Also Clear question, rationale is missing

Introduction: The problem generally is very relevant. Reforestation is global challenge, with developing trees being more succeptible to drought especially shortly after plantig compared to mature forests. The study summarizes the problem well giving an appropriate overview of existing literature. However, I fail to see what the novelty of their study is supposed to be in relation to already existing knowledge. If this is studying the root water uptake depth distributions, then they mus tailor their paper much more on the plasticity advantage during drought and why this might be a key strategiy also highlighting the knowledge gap more (to it seemed there is already some knowledge on this).

Specific: clear questions and hypotheses missing.

Material and Methods:

Are the tree differences (Table 1) significant between classes? ND and LD seem to be within error margin in some categories. Did you specify range in the different categories as target values for the different degradation stages or how was the evaluation procedure done?

I think this needs much more consideration: regarding degradation timing since planting, tree age and degradation intensity is very important, also why are the different spots that are in direct vicinity so different? What are differences in the soil or microclimate that might explain these? Targetting these question and conceptualizing this would greatly improve the work.

Isotope analysis: was cross referencing of the different analyzers performed? What technique did you use at the mass spec?

Results:

Figure 2: error margins in the lower panels of each plot?

Figure 4: harmonize the range on both axes. Colloring: care for problems with color blindness

Discussion:

The discussion is very much centered directly on the results in the beginning. It would be helpful to guide the reader again through the data and follow a read line revisiting central questions and hypothesis.

---

## Author Comment (AC1)

In line 320, the author concluded that the low transpiration in September resulted from the soil drought. I would like to inquire whether the authors have taken into account the possibility that transpiration declines in the late growing season, even with normal soil moisture availability.

**Reply**

5 We believed that it is unlikely that the transpiration of *Populus simonii* declines in the late growing season when the soil water availability is normal in this study area. The experimental plots were located on sandy land with >95% soil sand content in a semi-arid region. In the 2021 growing season, the average soil volumetric water content at 0-200 cm depth in the three plots was from 0.031 to 0.033 $cm^3$ $cm^{-3}$. As a fast-growing tree with high canopy transpiration demands, the growth of *P. simonii* was strictly restricted by soil water conditions.

10 To provide possible evidence, we calculated the transpiration of *P. simonii* for the 2022 growing season (May-September) (unpublished data). As shown in Fig S1 below, the soil water availability was not low (high REW) in each plot in September 2022. However total September transpiration of *P. simonii* in each plot was not reduced by the late growing season and even reached the highest in the ND and SD plots. In addition, the lowest transpiration of *P. simonii* in June 2022 was likely related to low soil water availability, with total precipitation in June of only 14.0 mm, much lower than the multi-year average of

15 52.2 mm (2003-2022). Moreover, in late June 2022, defoliation of *P. simonii* was observed in the three plots.

Besides, the transpiration of *P. simonii* was also affected by meteorological factors such as photosynthetically active radiation (PAR), vapor pressure deficit (VPD), and air temperature ($T_a$). In general, PAR, VPD, and $T_a$ decreased in the late growing season compared to July-August, which may somewhat affect the transpiration of *P. simonii*. Therefore, in the later revision, it may be more accurate for us to revise the statement here as "In other words, the high relative contribution

20 proportion of deep soil water during severe soil drought conditions did not increase the absolute water use. Meanwhile, severe soil drought decreased shallow and middle soil water use. Therefore, low soil water availability greatly limited *P. simonii* transpiration".

[Figure]

**Figure S1:** Monthly total transpiration and mean REW (relative extractable water) during 2021 (a) and 2022 (b) growing

25 seasons in the three plots. ND, LD, and SD indicate no, light, and severe degradation of *P. simonii*, respectively.

---

## Author Comment (AC2)

**Response to Reviewer #1, re: "Reduced transpiration without changes in root water uptake patterns in degraded trees in semi-arid afforestation ecosystems", in the review in HESSD (NO. hess-2023-66).**

**Reviewer #1:** Dai et al. present ecohydrological data of three sites varying in tree degradation with the general aim to shed light on potential differences in tree water cycling as cause for the degradation. The topic generally is very important, globally as well as in the studied region, and the data gathered are valuable to the scientific community. Before publication however, I strongly suggest major revisions to rework the paper and make the aim, outcome and implications much clearer.

**Reply**

We thank Reviewer #1 for thoughtfully and critically reviewing our manuscript. We greatly appreciate the detailed and thoughtful points that have certainly helped us to improve the manuscript. Overall, we agree with these suggestions and have made targeted amendments, as described in the detailed point-by-point replies to the Reviewer's comments below. The comments are cited in black. The response to each comment is presented in blue and passages changed in specific responses to the comments are presented in quotation marks and italic font.

**Specific comments**

**1. Title:** (Reduced transpiration without changes in root water uptake patterns in degraded trees in semi-arid afforestation ecosystems)

not exactly precise. The study wants to inform about afforestation systems about succeptability to drought. The title is too specific and also if that is the singular result it is not very novel or surprising.

**Reply**

This research contrasts in-depth the water-use strategies (transpiration, root water uptake pattern, and water-use efficiency) of trees under several degradation categories, and it highlights the potential significance of adjusting tree's water-use behavior for survival in sandy soil with restricted water availability. Additionally, the mechanism of interaction between tree degradation and soil water status was examined, providing a scientific foundation for plantation management in a semi-arid region. We think it would be more appropriate to change the original title to "*Changes in water-use strategies of degraded trees in semi-arid afforestation ecosystems*".

**2. Abstract:**

Stable isotope method: what is that supposed to mean? Sloppy formulation.

Hydrologic redistribution: How do you know? This is an interpretation based on what data?

**Reply**

Thanks for your comments. The following adjustments would allow us to more fully and completely communicate the research methodology and study content in the abstract:

"*To identify the changes in water-use strategies of degraded Populus simonii, the soil water content, hydrogen and oxygen isotopic compositions in the soil water and plant xylem water, carbon isotopic compositions in the leaf, and sap flow velocity of trees were continuously measured under various degradation degrees (no degradation, ND; light degradation, LD; or severe degradation, SD) during the 2021 growing season*".

General comment: The results are summarized in a very based and interpreted way and the concluding sentence does not really fit the described results. Also the different methods seem to contradict each other (e.g. different indication of root water uptake pattern from soil moisture compared to isotopic approaches). Also Clear question, rationale is missing.

**Reply**

To better fit the described results of our study, we intend to revise the concluding sentence as follows:

"*Our findings imply that degraded and no degraded trees in water-limited areas adopt substantially varied water-use strategies. The high-water requirements of no degraded trees may put deep soil water reservoirs at risk of depletion, leading to a conflict between forest and water*".

The findings of the isotope-based assessment of root water uptake patterns were not in conflict with soil moisture dynamics in our study. First of all, no degraded trees require more water than degraded trees (Fig. 6), and when shallow or middle soil water is insufficient to support their normal growth activities, they utilize more deep soil water use by deep root systems. However, deep water sources are difficult to replenish quickly, and precipitation may have been consumed by soil evaporation and transpiration from the vegetation before infiltrating into the deep layer (aside from specific heavy precipitation events). In the long run, deep soil water deficit worsens in the ND plot (Fig. 3). On the contrary, degraded trees have low water requirements (Fig. 6) and the SWC of the deep layer is higher in the LD and SD plots than that in the ND plot (Fig. 3). Secondly, based on the stable isotope mixing model, it was found that degraded trees also relied on a certain proportion of deep soil water during drought (Fig. 7), which is a way for the trees to respond to drought (Jiang et al., 2020; Wang et al., 2017). As a result of the fact that deep water sources are more stable and that degraded trees are still able to absorb deep soil water through their deep root systems (Fig. 2). However, due to low water requirements, degraded trees still used less deep soil water in absolute terms than no degraded trees, even when the proportion of deep water used by degraded trees was equivalent to that of no degraded trees (Fig. 7). We would like to make the following adjustments to clearly and logically display the results in the summary:

"*As tree degradation intensified, the root weight density decreased significantly ($P<0.05$) and the root proportion of the shallow layer (0–40 cm) increased. Influenced by precipitation recharge, the soil volumetric water content (SWC) of the shallow layer did not show significant differences ($P>0.05$) among the various degradation degrees. The SWC of the middle (40–80 cm) and deep (80–200 cm) layers were significantly ($P<0.05$) lower in the ND plot than the LD and SD plots, which was related to*

*the higher water demand by no degraded trees. Results from the stable isotope mixing model showed that during drought, degraded trees can still absorb deep soil water through their deep root systems as no degraded trees did. However, due to low water demand, degraded trees used less deep soil water in absolute terms than no degraded trees. Moreover, evidence from leaf carbon isotopes suggested that intrinsic water-use efficiency of degraded trees was more sensitive to SWC than no degraded trees*".

Wang, J., Fu, B., Lu, N., and Zhang, L.: Seasonal variation in water uptake patterns of three plant species based on stable isotopes in the semi-arid Loess Plateau, Sci. Total Environ., 609, 27–37, https://doi.org/10.1016/j.scitotenv.2019.02.262, 2017.

Jiang, P., Wang, H., Meinzer, F.C., Kou, L., Dai, X., Fu, X.: Linking reliance on deep soil water to resource economy strategies and abundance among coexisting understorey shrub species in subtropical pine plantations. New Phytol. 225, 222–233. https://doi.org/10.1111/nph.16027, 2020.

**3. Introduction:** The problem generally is very relevant. Reforestation is global challenge, with developing trees being more succeptible to drought especially shortly after planting compared to mature forests. The study summarizes the problem well giving an appropriate overview of existing literature. However, I fail to see what the novelty of their study is supposed to be in relation to already existing knowledge. If this is studying the root water uptake depth distributions, then they must tailor their paper much more on the plasticity advantage during drought and why this might be a key strategy also highlighting the knowledge gap more (to it seemed there is already some knowledge on this).

Specific: clear questions and hypotheses missing.

**Reply**

Reforestation is a global endeavor and an even more challenging task in areas with little precipitation. One has to take into account the vegetation carrying capacity of the local soil moisture and select appropriate species for afforestation. China has made impressive strides in afforestation over the past few decades, but an increasing number of studies have found that semi-arid regions are seeing widespread deterioration of planted trees, which are characterized by short height, small size, and

dieback (Chen et al., 2015; Liu et al., 2020; McVicar et al., 2010; Sun et al., 2018; Zhang et al., 2020). Undoubtedly, one of the key factors is the lack of water (Liu et al., 2021). To better understand how silvicultural species adapt to adversity and direct future silvicultural practices toward sustainable development of plantation forests, it is crucial to focus on how these degraded trees modify their water-use strategies.

There have been many studies in the past on the sources of water utilization in silvicultural vegetation, such as the comparison of the same tree species in mixed and pure forests (Tang et al., 2018), the comparison of different tree species in mixed forests (Grossiord et al., 2014; Liu et al., 2021b; Tang et al., 2019), different types of pure forests (Wang et al., 2020), and different tree ages (Wang et al., 2021). However, there is a lack of research comparing water-use strategies between degraded trees (organ structural changes and reduced function) and healthy trees. Filling these knowledge gaps will help clarify the mechanisms of tree degradation or mortality and provide a scientific basis for plantation forest management in semi-arid regions.

Due to their smaller size and narrower canopy, degraded trees may exhibit decreased transpiration rates. Additionally, because degraded trees lack deep root systems, they may be unable to utilize deep soil water (Liu et al., 2021a). Degraded trees with limited deep-root activity may be susceptible to drought since the availability of deep-water sources frequently permits trees to adopt isohydric behavior to reduce drought stress (Brinkmann et al., 2016; Ding et al., 2021). Thus, we hypothesized that with increased tree degradation, the deep roots and transpiration of *P. simonii* would decrease. This, in turn, would lead to a reduction in the relative contribution and absolute use of deep soil water and, as a result, a trend toward greater water storage in the deep layer.

To properly illustrate the potential innovations, scientific questions, research implications, and hypotheses of our work, we would like to make the following revisions to the Introduction:

*"Forests account for approximately 45% of global terrestrial carbon storage, play an essential role in the water cycle of terrestrial ecosystems, and provide various critical ecosystem services for maintaining biodiversity (Choat et al., 2018; Liu et al., 2021a). However, global warming has caused*

*more frequent regional drought events and increased the duration and intensity of droughts (Allen et al., 2010), especially in arid and semi-arid areas (Iqbal et al., 2021). These changes led to lower water availability and soil degradation in forests and reduced biodiversity in the affected areas, thereby reducing the forest area. To increase the carbon sink, enhance biodiversity, and prevent wind and soil erosion in water-limited areas, a series of afforestation ecological engineering projects have been implemented worldwide in recent decades (Chen et al., 2019; Yang et al., 2022), especially in China (Guo, 2021; Su and Shangguan, 2019). Although China only accounts for 6.6% of the vegetation area worldwide, it contributed 25% of the global net increase in leaf area from 2000 to 2017, thus contributing to global greening efforts (Chen et al., 2019). Reforestation is a challenging task in areas with little precipitation. One has to take into account the vegetation carrying capacity of the local soil moisture and select appropriate species for afforestation. Unfortunately, an increasing number of studies have found that semi-arid regions are seeing widespread deterioration of planted trees, which are characterized by short height, small size, and dieback (Chen et al., 2015; Liu et al., 2020; McVicar et al., 2010; Sun et al., 2018; Zhang et al., 2020). Undoubtedly, one of the key factors is the lack of water (Liu et al., 2021a). To better understand how silvicultural species adapt to adversity and direct future silvicultural practices toward sustainable development of plantation forests, it is crucial to focus on how these degraded trees modify their water-use strategies.*

*Compared to the functions of normal-growing trees, those of degraded trees declined significantly. Degraded trees might present a lower transpiration rate due to the small tree size and narrower canopy width, and they might also lose the ability to utilize deep soil water from the lack of deep root systems (Liu et al., 2021a). Because the availability of deep-water sources often enables trees to adopt isohydric behavior to mitigate drought stress (Brinkmann et al., 2016; Ding et al., 2021), degraded trees with low deep-root activity may be vulnerable to drought (Liu et al., 2021a). Currently, most studies suggest that tree degradation or mortality is associated with water stress (Liu et al., 2021a; Sun et al., 2018; Zhang et al., 2020). Under drought conditions, trees often suffer from reduced photosynthetic carbon sequestration and non-structural carbohydrate depletion (McDowell et al., 2008), which might stunt tree growth. As techniques for tracing stable isotopes ($^2H$, $^{18}O$, and $^{13}C$) have matured, the methods have been extensively used in critical eco-hydrology topics, such as determining the spatiotemporal*

sources of water taken up by plants (Miguez-Macho and Fan, 2021), calculating the mean transit time of various hydrological components (Dai et al., 2022), estimating the source water contribution to root water uptake (Dai et al., 2020), and analyzing the intrinsic water-use efficiency ($WUE_i$) of a plant (Wu et al., 2022). There have been many studies in the past on the sources of water utilization in silvicultural vegetation, such as the comparison of the same tree species in mixed and pure forests (Tang et al., 2018), the comparison of different tree species in mixed forests (Grossiord et al., 2014; Liu et al., 2021b; Tang et al., 2019), different types of pure forests (Wang et al., 2020), and different tree ages (Wang et al., 2021). However, there is a lack of research comparing water-use strategies between degraded trees and healthy trees. Filling these knowledge gaps will help clarify the mechanisms of tree degradation or mortality and provide a scientific basis for plantation forest management in semi-arid regions.

Generally, the species used in afforestation in water-limited areas have strong drought adaptability and can regulate water-use strategies under changing water conditions. For instance, drought-tolerant species can shift the water source from the shallow soil layer to the deep soil layer through dimorphic roots (Wang et al., 2017), increase the $WUE_i$ (Wang et al., 2021), or display more isohydric behavior to reduce water loss (Ding et al., 2021) under water stress conditions. Although seasonal patterns in deep soil water utilization help trees survive drought, they also accelerate the depletion of deep soil reservoirs, reducing the stability of plantations in water-limited areas (Chen et al., 2008; Su and Shangguan, 2019). Aside from certain heavy precipitation events, precipitation may have been consumed by soil evaporation and vegetation transpiration before infiltrating into the deep layer. Deep water sources are therefore difficult to replenish quickly. Notably, assessing the soil water-carrying capacity for vegetation in water-limited areas requires an understanding of the relative contributions of different water sources and absolute water consumption of afforestation species. We can calculate the whole tree or stand transpiration by using stable isotope technology, coupling the thermal dissipation method to monitor sap flow (Granier, 1987), and then we can comprehensively grasp the relationship between trees and water, preventing further plantation declines.

This study was conducted on Populus simonii plantations in a semi-arid area of the Chinese Loess Plateau. P. simonii is the local vanguard tree for afforestation projects; however, in recent years, the

*structure of trees in these plantations has changed and their function has declined in this area. First, we carried out a field investigation measuring tree growth indicators and stand structure parameters (height, size, canopy width, and dead branches) and then conducted a comprehensive evaluation to distinguish different degradation degrees of P. simonii. Then, the soil water content, hydrogen and oxygen isotopic compositions in the soil water and xylem water, carbon isotopic compositions in the leaf, and sap flow of trees were continuously measured at each sample site during the growing season. The objectives were to: (1) identify the isotopic characteristics of soil water, xylem water, and leaves, (2) compare the water-use strategies (transpiration, root water uptake patterns, and WUE$_i$) under different categories of degradation, and (3) elucidate the interaction mechanism between tree degradation and soil water status. We hypothesized that with increased tree degradation, the deep roots and transpiration of P. simonii will decrease, which will lead to a reduction in the relative contribution and absolute use of deep soil water and thus a trend of increasing water storage in the deep layer".*

**4. Material and Methods:**

Are the tree differences (Table 1) significant between classes? ND and LD seem to be within error margin in some categories. Did you specify range in the different categories as target values for the different degradation stages or how was the evaluation procedure done?

I think this needs much more consideration: regarding degradation timing since planting, tree age and degradation intensity is very important, also why are the different spots that are in direct vicinity so different? What are differences in the soil or microclimate that might explain these? Targeting these questions and conceptualizing this would greatly improve the work.

**Reply**

Concerning tree morphological indicators, we performed a one-way ANOVA and discovered that the primary distinctions between ND (no degradation) and LD (light degradation) were the spike top ratio and DBH (revised Table 1). The "spike top" was described as a dead area of the trunk in the upper

canopy of *P. simonii* trees showing continuous growth of lateral branches. The ND plot had no dead branches on the main trunk and the best tree growth out of the three plots. However, the SD plot had the worst tree growth in the three plots and had a high spike top ratio. In our study, we classified the level of tree degradation based mostly on the magnitude of the spike top ratio. According to the criteria employed in earlier studies (Liu et al., 2020; Liu et al., 2021a), we defined a tree with a spike top ratio of 0–40% as LD, whereas a spike top ratio of more than 40% as SD.

Thanks for your helpful suggestions. They are interesting to think about. Firstly, the study's target trees were planted during the 1999 reforestation effort. The timing of tree degradation is difficult to monitor because of the substantial labor and time costs to the researcher, which was regrettably not possible to do in our study. Secondly, as mentioned in the previous paragraph, it may be appropriate to assess the degree of degradation in terms of the spike top ratio in our study. Our early field investigation revealed that the three nearby sample sites had a similar microtopography, climate, and soil texture (soil granules >95% sand). Because the objective of this study was to examine the water-use strategies of trees with various growth status (healthy and unhealthy), we focused on the sample trees with notable differences in growth indicators. After more than 20 years of growth under natural settings, intraspecific variation in the growth status of the same planted trees is not odd, and the specific causes for this are complex. For instance, trees are extremely sensitive to environmental stresses and prone to pests and diseases in the early stages after planting, and 100% survival cannot be guaranteed. It is also not possible to ensure that all saplings receive the same treatments at planting, e.g., differences in the amount of irrigation water. In summary, we intend to reveal the potential causes of tree deterioration from the perspective of their hydro-ecological behavior by comparing the water-use strategies of trees with different categories of degradation.

We would like to conduct a one-way ANOVA on the tree morphological indicators for each of the sampled sites in Table 1 to clearly represent the differences in tree growth status (revised Table 1). Specific criteria would be created, and the quantitative indicators for assessing the categories of tree degradation in terms of the magnitude of the spike top ratio will be defined. And, other growth

indicators would be supplemented with qualitative assessment. Some modifications in section 2.2 of "determination of sampling sites to reflect different categories of degradation" are as follows:

*"First, we determined three 50 × 50 m² plots with distinct growth differences in height, size, canopy width, and spike top (Fig. 1b). Their microtopography, climate, and soil texture were similar. P. simonii in the three plots experienced varying degrees of degradation after years of growth. Then, four 20 × 20 m² quadrats were set up in each plot, and the growth indicators of all P. simonii were measured, including the tree height, spike top height, diameter at breast height (DBH), and canopy projection area ($A_c$, calculated by the crown width in the east-west and north-south directions). The canopy density of each quadrat was obtained from the ratio of total $A_c$ to total area. We surveyed 128 trees in each of the sample plots. The basic information above is shown in Table 1. Notably, in one of the three plots, P. simonii had the best growth according to fine growth indicators and no dead branches on the main trunk (Fig. 1c), and this plot was defined as no degradation (ND). The mean height, DBH, and $A_c$ of trees in the ND plot were 10.3 m, 19.2 cm, and 10.3 m², respectively. However, the growth status of P. simonii in the other two plots was inferior to that in the ND plot to varying degrees, and the P. simonii trees exhibited a spike top. According to the criteria employed in earlier studies (Liu et al., 2020; Liu et al., 2021a), we defined a tree with a spike top ratio of 0–40% as light degradation (LD), whereas spike top ratio of more than 40% as severe degradation (SD). The spike top ratio was derived by dividing the average spike top height by the average tree height. The DBH and spike top ratio of P. simonii in the LD plot were significantly (P<0.05) smaller than those in the ND plot (Table 1). P. simonii in the SD plot had the worst growth parameters, with an average height, DBH, and $A_c$ of 5.9 m, 12.5 cm, and 5.9 m², respectively (Table 1)".*

Table 1  Basic characteristics of *P. simonii* at different sampling sites

| Growth status | Tree height (m) | Spike top height (m) | Spike top ratio (%) | DBH (cm) | $A_c$ (m²) | Crown density (%) |
|---|---|---|---|---|---|---|
| ND | 10.3 ± 1.3a | 0a | 0a | 19.2 ± 3.8a | 10.3 ± 3.4a | 83.4 ± 13.2a |
| LD | 9.3 ± 1.0a | 2.6 ± 0.4b | 28.0 ± 3.5b | 15.4 ± 2.1b | 9.0 ± 1.9a | 72.0 ± 9.5a |
| SD | 5.9 ± 0.8b | 2.7 ± 0.3b | 45.8 ± 3.1c | 12.5 ± 1.6c | 5.9 ± 1.5b | 47.2 ± 8.4b |

The DBH and $A_c$ represent diameter at breast height and canopy projection area, respectively. Values are shown as the mean ± standard deviation. Different lowercase letters indicate significant differences in tree morphological indicators among different sampling sites (*P*<0.05), based on the Tukey HSD post hoc test in One-way ANOVA.

Liu, Z., Jia, G., and Yu, X.: Variation of water uptake in degradation agroforestry shelterbelts on the North China Plain, Agr. Ecosyst. Environ., 287, 106697, https://doi.org/10.1016/j.agee.2019.106697, 2020.

Liu, Z., Jia, G., Yu, X., Lu, W., Sun, L., Wang, Y., and Zierdie, B.: Morphological trait as a determining factor for *Populus simonii* Carr. to survive from drought in semi-arid region, Agr. Water Manage., 253, 106943, https://doi.org/10.1016/j.agwat.2021.106943, 2021a.

Isotope analysis: was cross referencing of the different analyzers performed? What technique did you use at the mass spec?

**Reply**

We used equipment from the Institute of Water-saving Agriculture Research in Chinese Arid Areas, Northwest A&F University for the liquid-water isotopic analyzer (912-0032, LGR Inc., California, USA) and stable isotope ratio mass spectrometer (Isoprime-100, Isoprime Inc., Cheadle, UK). The $\delta^2$H and $\delta^{18}$O in the water determined by the two devices were compared by the laboratory administrators. The two instruments' accuracy ($\delta^2$H: 0.5‰ and $\delta^{18}$O: 0.1‰) was met by their test findings, and there was no difference between test findings (Li et al., 2020). Organic matter in plant xylem water will interfere with the analysis results of the liquid-water isotopic analyzer (Schultz et al., 2011), but will not affect the test results of the mass spectrometer (Li et al., 2020). In addition, the test cost of the liquid-water isotopic analyzer is less than that of the mass spectrometer. Therefore, we measured the water isotopes using a mass spectrometer. As a result, we measured water isotopes in the plant xylem using a mass spectrometer and other water samples using a liquid-water isotopic analyzer.

Two modes exist in the mass spectrometer. In soil and plant samples, solid testing mode is used to determine the $\delta^{13}$C, $\delta^{15}$N, and total C and N; liquid testing mode is used to determine the $\delta^2$H and $\delta^{18}$O in water samples. The laboratory administrators conduct tests on each of our samples in compliance with the IsoPrime mass spectrometer manual supplied by Isoprime Ltd.

We would like to add some information about isotope analysis as shown below:

*"The hydrogen and oxygen isotopic compositions of xylem water were analyzed using a stable isotope ratio mass spectrometer (Isoprime-100, Isoprime Inc., Cheadle, UK) to avoid spectral pollution. Cross-testing the same set of LGR4E standard water samples (manufactured by LGR Inc., USA) with the two devices yielded identical results. The carbon isotopic compositions of leaf samples were analyzed using the stable isotope ratio mass spectrometer (solid sample testing mode)".*

Schultz, N.M., Griffis, T.J., Lee, X., Baker, J.M.: Identification and correction of spectral contamination in $^2H/^1H$ and $^{18}O/^{16}O$ measured in leaf, stem, and soil water. Rapid Commun. Mass Spectrom. 25, 3360–3368. https://doi.org/10.1002/rcm.5236, 2011.

Li, H., Jin, J., Nie, J., Yang, B., Xu, Y., Ding, R.: Determination of $\delta^2H$ and $\delta^{18}O$ values in water by element analyzer-isotope ratio mass spectrometer. Analytical Instrumentation 6, 106–110. https://doi.org/10.3969/j.issn.1001-232x.2020.06.020., 2020. (In Chinese)

**5. Results:**

Figure 2: error margins in the lower panels of each plot?

Figure 4: harmonize the range on both axes. Colloring: care for problems with color blindness.

**Reply**

Thanks for your comments. We have revised Fig. 2 and Fig. 4 as follows:

[Figure]

Figure 2: Horizontal and vertical root distributions of *P. simonii*. The box plots show the minimum, 25% percent, median, 75% percent, and maximum of root weight density at 50, 100, and 200 cm from the trunk in the 0–200 cm soil profile, and black squares indicate the mean. The inset shows a properly enlarged vertical root distribution on the X-axis. Error bars show the standard deviation.

[Figure]

Figure 4: Scatter distributions of $\delta^2H$ and $\delta^{18}O$ in precipitation, soil water, and xylem water. LMWL represents the local meteoric water line ($\delta^2H=7.17\delta^{18}O+3.19$, $R^2=0.97$), and GMWL represents the global meteoric water line ($\delta^2H=8\delta^{18}O+10$). $SW_{0-40}$, $SW_{40-80}$, and $SW_{80-200}$ represent the soil water in shallow (0–40 cm), middle (40–80 cm), and deep (80–200 cm) layers, respectively.

**6. Discussion:** The discussion is very much centered directly on the results in the beginning. It would be helpful to guide the reader again through the data and follow a read line revisiting central questions and hypothesis.

**Reply**

We concur with you. The three sections of the Discussion correspond to the three research objectives mentioned in the Introduction. In Discussion 4.1, we analyzed the characteristics of isotopic compositions in different water bodies; these data results were a key part of the study of plant water-use sources using isotope mixing models and would be of interest to scholars in the field of isotope hydrology. In Discussion 4.2, we analyzed the changes in water-use strategies of degraded trees compared to no degraded trees in terms of root distribution, transpiration, root water uptake patterns, and $WUE_i$, and reviewed the hypotheses of this paper. In Discussion 4.3, we have elucidated the interaction mechanism between tree degradation and soil water status according to the main findings of the article and have proposed some measures that can optimize afforestation in moisture-limited areas, which helps to clarify the mechanism of tree degradation or mortality and provides a scientific basis for the management of planted forests in semi-arid zones.

We would like to modify part of Discussion 4.2 to review the main findings and hypotheses, as follows:

*"As degradation intensified, tree transpiration significantly decreased (P<0.05). The higher the degradation degree, the lower the DBH, crown width, and root biomass of the tree (Fig. 1 and Fig. 2), and the less water needed for tree growth. Reduced water demand of degraded trees facilitated deep soil moisture storage (Fig. 3). However, the root water uptake patterns were similar in terms of both time and depth in the ND, LD, and SD plots (Fig. 7a). Thus, our findings rejected the original hypothesis regarding the root water use patterns while supported the other hypotheses. In our study, the ability of degraded trees to absorb deep soil water had not changed, which was inconsistent with that of Liu et al. (2020) and Liu et al. (2021a)".*

---

## Author Comment (AC3)

**Response to Reviewer #2, re: "Reduced transpiration without changes in root water uptake patterns in degraded trees in semi-arid afforestation ecosystems", in the review in HESSD (NO. hess-2023-66).**

We thank Reviewer #2 for thoughtfully and critically reviewing our manuscript. We greatly appreciate the detailed and thoughtful points that have certainly helped us to improve the manuscript. Overall, we agree with these suggestions and have made targeted amendments, as described in the detailed point-by-point replies to the Reviewer's comments below. The comments are cited in black. The response to each comment is presented in blue and passages changed in specific responses to the comments are presented in quotation marks and italic font.

**Reviewer #2:** The focus of this manuscript is the evaluation of the water-use strategies of *Populus simonii* trees under different degradation conditions. The authors combined the analysis of the root system, the isotopic composition (hydrogen and oxygen) of xylem and soil water, soil water content and sap flow to test whether degraded trees used less deep soil water compared to non-degraded trees.

The topic of this manuscript is potentially interesting for the readers of Hydrology and Earth System Sciences, and the paper is generally well written, even though I recommend a revision of the English by a native speaker. Besides this, I have some comments that would require a major revision of the text. First, the authors should clearly present the general objective and novelty of their work (this should be emphasized in the discussion and the conclusions as well). Secondly, the authors should specify some more methodological details and discuss the limitations of their study (please see the specific comments). Thirdly, the authors should consider restructuring the results, by starting from the presentation of the physical characteristics of the trees and the root systems, then by describing soil moisture and sap flow dynamics, and finally by presenting the quantification of the contribution of soil water at different depths to xylem water.

**Reply**

Before submitting this article, we asked the agency of Wiley Editing Services for English polishing services. We apologize that the previous edit was unsatisfactory. We're looking for a native speaker to check the English again. The specific changes will appear in the revised manuscript.

We will add a conceptual figure to the Discussion section to present the general objective and novelty of the work. See response to specific comment 6 for details. Some modifications are as follows:

*"To clarify the mechanisms of tree degradation or mortality and direct future silvicultural practices toward sustainable development of plantations, it is critical to be aware of how these degraded trees modify their water-use strategies (transpiration, root water uptake patterns, and $WUE_i$). As summarized in Fig. 9, tree transpiration significantly decreased with intensified degradation, ($P<0.05$). The higher the degradation degree, the lower the DBH, crown width, and root biomass of the tree (Fig. 1, Fig. 2, and Fig. 9), and the less water needed for tree growth. Reduced water demand of degraded trees facilitated deep soil moisture storage (Fig. 3 and Fig. 9). These findings support our original hypothesis. However, the hypothesis that the relative contribution of deep soil water reduces with increased tree degradation was rejected. The root water uptake patterns were similar in terms of both time and depth in the ND, LD, and SD sites (Fig. 7a and Fig. 9)"* in Discussion section 4.2.

*"In conclusion, our findings imply that degraded and no degraded trees in water-limited areas adopt substantially varied water-use strategies. No degraded trees can utilize more deep soil water storage through a developed deep root system during the dry period; degraded trees can enhance their drought adaptability by improving $WUE_i$ under soil water stress. However, the high water demand of artificial arboreal forests under normal conditions may threaten deep soil water reservoirs. Therefore, stand management, including appropriate thinning and mixed tree-shrub patterns, may be necessary to improve soil moisture conditions"* in the Conclusions section.

We will specify some more methodological details and discuss the limitations of our study. See response to specific comments 1, 2, 3, and 5 for details.

We will restructure the Results section. See response to specific comment 4 for details.

**Specific comments**

1. Section 2.3: The authors should clearly report in the text the number of samples (soil and plants) collected for isotopic analyses in each site and for each sampling date. To improve the clarity of the results, sample size (n) should be always reported in tables (e.g., Tables 1 and 2) and figures (or in their caption), and in the presentation of the results of statistical analyses.

**Reply**

To clearly report the number of samples (soil and plants), we intend to revise the sentence as follows:

"*We collected three xylem and leaf samples at each site and on each sampling date, and a total of 45 xylem samples and 45 leaf samples were collected during the observation period*".

"*The sampling depth was 0–200 cm, and the sampling intervals were 10 cm and 20 cm at the depth of 0–20 cm and below 20 cm, respectively. We collected 33 soil samples at each site and on each sampling date, and a total of 495 soil samples were collected during the observation period*".

To improve the clarity of the results, we will report the sample size (*n*) in tables and figures (or their captions) as well as in the results of the statistical analysis, with some modifications as follows:

Table 1  Basic characteristics of *P. simonii* at different sampling sites

| Growth status | Tree height (m) | Spike top height (m) | Spike top ratio (%) | DBH (cm) | $A_c$ (m²) | Crown density (%) |
|---|---|---|---|---|---|---|
| ND | 10.3 ± 1.3a | 0a | 0a | 19.2 ± 3.8a | 10.3 ± 3.4a | 83.4 ± 13.2a |
| LD | 9.3 ± 1.0a | 2.6 ± 0.4b | 28.0 ± 3.5b | 15.4 ± 2.1b | 9.0 ± 1.9a | 72.0 ± 9.5a |
| SD | 5.9 ± 0.8b | 2.7 ± 0.3b | 45.8 ± 3.1c | 12.5 ± 1.6c | 5.9 ± 1.5b | 47.2 ± 8.4b |

The DBH and $A_c$ represent diameter at breast height and canopy projection area, respectively. Values are shown as the mean ± standard deviation. The number of samples (*n*) was 128 for the tree growth indicators, except for canopy density (*n*=4). Different lowercase letters indicate significant differences in tree morphological indicators among different sampling sites (*P*<0.05), based on the Tukey HSD post hoc test in One-way ANOVA.

[revised manuscript text omitted]

2. Line 157: More quantitative details about rainfall events (amounts, intensity, and duration), and how they were defined (time without rainfall) are needed.

**Reply**

In our study, we only collected daily-scale precipitation samples and precipitation amounts. Notably, each rainfall event could not have more than 4 h of consecutive precipitation-free periods. If there were multiple rainfall events in one day, their samples were combined into one precipitation sample. We will add some detailed precipitation sample information as follows:

"*Daily-scale precipitation samples and amount were obtained from May to September 2021 (n=15) using a rain gauge placed on the open ground ~1000 m away from the P. simonii sampling sites. Because light rainfall events (precipitation <3 mm in one event) only moistened a few centimeters of soil in the P. simonii plantation, we collected precipitation samples in 30-mL polyethylene bottles after each rainfall event with precipitation >3 mm, immediately sealed them with parafilm, and transported them to the laboratory for frozen storage. Notably, each rainfall event could not have more than 4 h of consecutive precipitation-free periods. If there were multiple rainfall events in one day, their samples were combined into one precipitation sample*".

3. Section 2.5: The authors should clarify whether the thermal dissipation probes were calibrated or not, and add the measurement uncertainty, as well as the estimation of uncertainty in terms of daily transpiration. Furthermore, it is unclear why the authors monitored sap flow in five trees that were not used for the collection of xylem water (this should be justified). What are the main physical characteristics of the trees chosen for isotopic sampling and how did they differ from the trees selected for sap flow monitoring?

**Reply**

We did not perform laboratory calibration on the newly purchased thermal dissipation probes, because several studies have found that laboratory calibration of sap flow for diffuse-porous species is not significantly different from Granier's empirical calibration (Granier, 1987). Using Granier's empirically calibrated formula can estimate transpiration in diffuse-porous trees but underestimate transpiration in ring-porous wood trees (Taneda and Sperry 2008; Bush et al. 2010; Ma, 2018). In this study, *P. simonii* belongs to diffuse-porous species (Dai et al., 2020).

Uncertainties in the measurement of sap flow using thermal dissipation probes arise mainly from differences in sap flow radial transport velocity in the sapwood and differences in sap flow densities between different orientations of the same tree. These uncertainties in sap flow measurements may lead to uncertainty in transpiration estimation. The objective of this study was to compare the transpiration and root water uptake strategies and their response to drought in *P. simonii* with different degrees of degradation. We used the same standard sensor to measure the sap flow velocity of the same species and used the same methodology to estimate the daily transpiration at the whole-tree scale. Therefore, the uncertainties in measurement and transpiration estimation can be ignored in this study.

A portion of the branches of the sample tree was cut each month when plant samples (xylem and leaves) were collected. The selection of these sample trees for sap flow measurements may affect the results of the observations. The morphological indicators of the sample trees used for isotopic sampling and sap flow monitoring were both close to the average of trees of each site in Table 1.

We will make the following adjustments to specify some more methodological details:

"*Notably, we did not perform laboratory calibration on the TDP-20, because several studies have found that laboratory calibration of sap flow for diffuse-porous species (i.e. P. simonii) is not significantly different from Granier's empirical calibration (Bush et al. 2010; Dai et al., 2020b; Taneda and Sperry 2008). Uncertainties in the measurement of sap flow using TDP-20 arise mainly from differences in sap flow radial transport velocity in the sapwood and differences in sap flow densities between different orientations of the same tree. These uncertainties in sap flow measurements may lead to uncertainty in transpiration estimation. Because we used the same standard sensor to measure the sap flow velocity of*

*the same species and used the same methodology to estimate the daily transpiration at the whole-tree scale, the uncertainties in measurement and transpiration estimation can be ignored in this study comparing water-use strategies of P. simonii with different levels of degradation*".

"*We selected three sample trees at each fixed site as well as three soil profiles for isotopic sampling, and these sample trees had similar morphological indicators to the average of trees of each site in Table 1*".

"*To avoid the influence of the removal of branches on the sap flow measurement during isotopic sampling, at each sample site (ND, LD, and SD), the sap flow velocity of five trees close to the soil and plant sampling sites was continuously monitored using a thermal dissipation probe (TDP-20, Dynamax Inc., TX, USA) at 1.3 m above the ground on the north side from June to September in 2021 (P. simonii leaves fall extensively at the beginning of October). Data were recorded by loggers (CR-1000, Campbell Scientific, Utah, USA) in 30 min intervals. The height, DBH, and $A_c$ of these sample trees were close to the mean of each sample site in Table 1*".

Granier, A.: Evaluation of transpiration in a Douglas-fir stand by means of sap flow measurements. Tree Physiol., 3, 309–320, https://doi.org/10.1093/treephys/3.4.309, 1987.

Taneda, H., and Sperry, J. S.: A case-study of water transport in co-occurring ring- versus diffuse-porous trees: contrasts in water-status, conducting capacity, cavitation and vessel refilling. Tree Physiol., 28, 1641–1651, https://doi.org/10.1093/treephys/28.11.1641, 2008.

Bush, S. E., Hultine. K. R., Sperry, J. S., and Ehleringer, J. R.: Calibration of thermal dissipation sap flow probes for ring- and diffuse-porous trees. Tree Physiol., 30, 1545–1554, https://doi.org/10.1093/treephys/tpq096, 2010.

Ma, C. K.: Study on the ecohydrological processes of black locust forest land on the loess plateau. Dissertation for Doctor Degree, Northwest A&F University, 2018. (in Chinese)

Dai, Y. X., Wang, L., and Wan, X. C.: Frost fatigue and its spring recovery of xylem conduits in ring-porous, diffuse-porous, and coniferous species in situ. Plant Physiol. Bioch., 146, 177–186, https://doi.org/10.1016/j.plaphy.2019.11.014, 2020.

4. Lines 241-250: This text can be moved to Section 3.1 or to a new section about the dynamics of soil water content.

**Reply**

We accept the reviewer's suggestion and will restructure the Results section. Firstly, we present the horizontal and vertical root distributions of *P. simonii* at each site, which are crucial physical characteristics of root water uptake of trees. The morphological indicators of the sample trees are presented in Table 1 of Section 2.2. Secondly, we describe soil moisture and transpiration dynamics in Section 3.2. Then, we characterize isotopic compositions of different water pools, which is a key part of the study of plant water-use sources using isotope mixing models and will be of interest to scholars in the field of isotope hydrology. Finally, we describe the quantification of the contribution of soil water at different depths to xylem water. Specific changes will appear in the revised manuscript.

5. Section 4: This section lacks some paragraphs about the main limitations of this study. For instance, the authors have not discussed how the methodological uncertainty in the extraction of xylem and soil water (by cryogenic vacuum distillation) might have affected their results and interpretation, especially in terms of the estimated contribution of soil water at different depths to xylem water.

**Reply**

Uncertainty in the extraction methodology of xylem and soil water may affect the results of isotope analysis and the calculation of the contribution proportion of soil water to xylem water at different depths. The most widely used approach for determining the $^2$H and $^{18}$O in plant and soil water is by first extracting water from soils and plants using the cryogenic vacuum distillation (CVD) method (Orlowski et al., 2018). Recent studies have challenged the interpretation of plant water isotopes obtained through CVD based on observations of a large $^2$H fractionation. Based on a rehydration experiment, Chen et al. (2020) believed that the xylem water cryogenic extraction error could originate from a dynamic exchange between organically bound deuterium and liquid water during water extraction. However, Diao et al. (2022) observed that $^2$H fractionation has an inversely proportional relationship with the

absolute amount of water being extracted and the methodological uncertainties can be controlled when sufficiently high amounts of xylem water were extracted (>0.6 mL). In our study, all xylem water samples obtained through CVD were about 1.0 mL and the isotope fractionation during CVD extraction of water can be negligible. In addition, some studies have also questioned the CVD accuracy on the extraction of soil water, because the evidence that cryogenically extracted soil water is depleted compared to reference water was found. Wen et al. (2021) and Yang et al. (2023) found that the cryogenic extraction biases were positively correlated with soil clay content. Orlowski et al. (2018) believed that sandy soil was almost unaffected by cryogenic extraction biases. In our study, the soil texture at each site is sandy (USDA classification), and the granular composition is 95.7% sand, 3.2% silt, and 1.1% clay. Moreover, no isotopic offset between soil water and xylem water was found (Fig. 4). Overall, the methodological uncertainty in the extraction of xylem and soil water (by CVD) can be negligible in this study.

We will add the discussion of the methodological uncertainty in the extraction of xylem and soil water (by CVD) to Section 4.1, as follows:

*"Uncertainty in the extraction methodology of xylem and soil water may affect the results of isotope analysis and the calculation of the contribution proportion of soil water to xylem water at different depths. The cryogenic vacuum distillation (CVD) method is widely used to extract water from soil and plant xylem for isotope analyses (Orlowski et al., 2018). Based on observations of a significant $^2H$ fractionation, recent investigations have questioned the interpretation of plant water isotopes obtained through CVD. Chen et al. (2020) postulated that the xylem water cryogenic extraction error may arise from a dynamic exchange between organically bound deuterium and liquid water during water extraction based on a rehydration experiment. However, Diao et al. (2022) observed that $^2H$ fractionation has an inversely proportional relationship with the absolute amount of water being extracted and the methodological uncertainties can be controlled when sufficiently high amounts of xylem water were extracted (>0.6 mL). In our study, all xylem water samples obtained through CVD were about 1.0 mL and the isotope fractionation during CVD extraction of water can be negligible. Due to evidence showing that cryogenically extracted soil water is depleted relative to reference water, some*

*researchers have also questioned the CVD's accuracy in extracting soil water. Wen et al. (2021) and Yang et al. (2023) found that the cryogenic extraction biases were positively correlated with soil clay content. Orlowski et al. (2018) believed that sandy soil was almost unaffected by cryogenic extraction biases. In our study, the soil texture at each site is sandy (USDA classification), and the granular composition is 95.7% sand, 3.2% silt, and 1.1% clay. Moreover, no isotopic offset between soil water and xylem water was found in Fig. 4. Overall, the methodological uncertainty in the extraction of xylem and soil water (by CVD) can be negligible in this study*".

Orlowski, N., Winkler, A., McDonnell, J. J., and Breuer, L.: A simple greenhouse experiment to explore the effect of cryogenic water extraction for tracing plant source water. Ecohydrology, 11, e1967, https://doi.org/10.1002/eco.1967, 2018.

Chen, Y. L., Helliker, B. R., Tang, X. H., Li, F., Zhou, Y. P., and Song, X.: Stem water cryogenic extraction biases estimation in deuterium isotope composition of plant source water. P. Natl. Acad. Sci. USA, 118, 33345–33350, https://doi.org/10.1073/pnas.20144 22117, 2020.

Wen, M., Lu, Y., Li, M., He, D., Wei, X., Zhao, Y., Cui, B., and Si, B.: Correction of cryogenic vacuum extraction biases and potential effects on soil water isotopes application. J. Hydrol., 603, 127011, https://doi.org/10.1016/j.jhydrol.2021.127011, 2021.

Diao, H., Schuler, P., Goldsmith, G. R., Siegwolf, R. T. W., Saurer, M., and Lehmann, M. M.: Technical note: On uncertainties in plant water isotopic composition following extraction by cryogenic vacuum distillation. Hydrol. Earth Syst. Sci., 26, 5835–5847, https://doi.org/10.5194/hess-26-5835-2022, 2022.

Yang, B., Dossa, G. G. O., Hu, Y. H., Liu, L. L., Meng, X. J., Du, Y. Y., Li, J. Y., Zhu, X. A., Zhang, Y. J., Singh, A. K., Yuan, X., Wu, J. E., Zakari, S., Liu, W. J., and Song, L.: Uncorrected soil water isotopes through cryogenic vacuum distillation may lead to a false estimation on plant water sources. Methods Ecol. Evol., 14, 1443–1456, https://doi.org/10.1111/2041-210X.14107, 2023.

**6.** The main findings of this study could be presented in a conceptual model and an accompanying figure/sketch.

**Reply**

We accept the reviewer's suggestion and will add a conceptual figure, as follows:

*"As summarized in Fig. 9, tree transpiration significantly decreased with intensified degradation, (P<0.05). The higher the degradation degree, the lower the DBH, crown width, and root biomass of the tree (Fig. 1, Fig. 2, and Fig. 9), and the less water needed for tree growth. Reduced water demand of degraded trees facilitated deep soil moisture storage (Fig. 3 and Fig. 9). These findings support our original hypothesis. However, the hypothesis that the relative contribution of deep soil water reduces with increased tree degradation was rejected. The root water uptake patterns were similar in terms of both time and depth in the ND, LD, and SD sites (Fig. 7a and Fig. 9)".*

[Figure]

Figure 9: Graphical summary of tree morphological characteristics, soil water status, and water-use strategies under different degradation degrees of *P. simonii* during the growing period (May to September). The soil water content (SWC) and transpiration (Tr) in the figure represent the mean values

for each site during the growing period. The data in the solid and dashed boxes represent the average proportion of soil water used by trees in May (the wettest month) and September (the driest month), respectively. As the SWC decreases, the intrinsic water-use efficiency ($WUE_i$) in the no degradation site stabilizes, and the $WUE_i$ in the light and severe degradation sites tends to increase.

7. Lines 435-436: I recommend reporting the description of the hypotheses here as well.

**Reply**

We accept the reviewer's suggestion and will report the description of the hypotheses here as follows:

"*Thus, our findings supported the hypothesis that the deep roots and transpiration of P. simonii decrease with increased tree degradation, which leads to a reduction in absolute use of deep soil water and thus a trend of increasing water storage in the deep layer. However, the hypothesis that the relative contribution of deep soil water reduces with increased tree degradation was rejected*".

**Technical corrections**

1. Line 6: Please delete 'method' and use 'stable isotopes'.

**Reply**

We agree with it. We will delete 'method' and use 'stable isotope'. The adjusted sentence is as follows:

"*To identify the changes in water-use strategies of degraded Populus simonii, the soil water content, hydrogen and oxygen isotopic compositions in the soil water and plant xylem water, carbon isotopic compositions in the leaf, and sap flow velocity of trees were continuously measured under various degradation degrees (no degradation, ND; light degradation, LD; or severe degradation, SD) during the 2021 growing season*".

2. Line 57: Techniques do not mature; I suggest rephrasing the current unclear sentence.

**Reply**

We accept the reviewer's suggestion and will rephrase the current unclear sentence, as follows:

*"Recently, stable isotopes ($^2H$, $^{18}O$, and $^{13}C$) have been extensively used in critical eco-hydrology topics, such as determining the spatiotemporal sources of water taken up by plants (Miguez-Macho and Fan, 2021), calculating the mean transit time of various hydrological components (Dai et al., 2022), estimating the source water contribution to root water uptake (Dai et al., 2020), and analyzing the intrinsic water-use efficiency (WUE$_i$) of a plant (Wu et al., 2022)".*

3. Line 70: Please delete 'technology' and use 'stable isotopes'.

**Reply**

We will amend the sentence as suggested:

*"We can calculate the water consumption from soil water storage by coupling stable isotopes and thermal dissipation method (Granier, 1987), and then we can comprehensively grasp the relationship between trees and water, preventing further plantation declines".*

4. Line 111: Please use 'plots' instead of 'quadrats'.

**Reply**

We will revise it in the next version as follows:

*"Then, four 20 × 20 m$^2$ plots were set up in each site, and the growth indicators of all P. simonii were measured, including the tree height, spike top height, diameter at breast height (DBH), and canopy projection area (Ac, calculated by the crown width in the east-west and north-south directions). The canopy density of each site was obtained from the ratio of total A$_c$ to total area".*

5. Line 118: Please replace 'inferior' with 'lower'.

**Reply**

We will revise it in the next version as follows:

*"However, the growth status of P. simonii in the other two sites was lower than that in the ND site to varying degrees, and the P. simonii trees exhibited a spike top".*

6. Line 147: Please replace 'rate' with 'efficiency'.

**Reply**

We will amend it as suggested:

"*A cryogenic vacuum distillation system (Li-2100, LICA Inc., Beijing, China) was employed to extract water from the xylem and soil, and the water extraction efficiency was over 99%*".

7. Lines 256 and 259: Please replace 'scatter points' with another term.

**Reply**

We will change it to "data points", as follows:

"*With increasing depth, the data points of $\delta^2H$-$\delta^{18}O$ for soil water gradually approached and clustered toward the LMWL in Fig. 4. The relationships of $\delta^2H$-$\delta^{18}O$ for soil water (soil water evaporation line, SWL) in the ND, LD, and SD sites showed similar characteristics (ND: $\delta^2H=4.29\delta^{18}O–33.21$, $R^2=0.81$; LD: $\delta^2H=4.29\delta^{18}O–34.59$, $R^2=0.87$; SD: $\delta^2H=4.05\delta^{18}O–35.77$, $R^2=0.82$). The data points of $\delta^2H$-$\delta^{18}O$ for xylem water matched those for soil water in the dual-isotope space (Fig. 4), thus implying a lack of apparent deuterium depletion*".

[Figure]

Figure 4: Relationships between $\delta^2H$ and $\delta^{18}O$ for precipitation, soil water, and xylem water. LMWL represents the local meteoric water line ($\delta^2H=7.17\delta^{18}O+3.19$, $R^2=0.97$), and GMWL represents the global meteoric water line ($\delta^2H=8\delta^{18}O+10$). $SW_{0-40}$, $SW_{40-80}$, and $SW_{80-200}$ represent the soil water in shallow (0–40 cm), middle (40–80 cm), and deep (80–200 cm) layers, respectively.

8. Line 304: Please replace 'middle' with the specific soil depth.

**Reply**

We agree with it and will replace 'middle' with the specific soil depth, as follows:

*"P. simonii in the ND and LD sites mainly absorbed shallow soil water in May (the month when the soil was the wettest), with contributions of 63.1% and 61.7%, respectively, while P. simonii in the SD site mainly absorbed middle soil water at the depth of 40–80 cm (48.0%) in May (Fig. 7a, b). P. simonii in the ND, LD, and SD sites mainly absorbed soil water at the depth of 40–80 cm from June to August, with contribution proportions of 42.1–45.0%, 38.3–45.2%, and 42.4–47.7%, respectively (Fig. 7a)".*